# Inference, Fast and Slow: Reinterpreting VAEs for OOD Detection

## Abstract

Unsupervised out-of-distribution (OOD) detection is critical for the safe deployment of machine learning systems, yet standard likelihood-based methods using deep generative models (DGMs) often fail, assigning deceptively high likelihoods to anomalous data. We attribute this failure, particularly within Variational Autoencoders (VAEs), to a phenomenon we term *likelihood cancellation*: informative signals from the model's encoder and decoder can neutralize each other within the final scalar likelihood. To overcome this, we introduce the **Likelihood Path (LPath) Principle**, a new framework that extracts a robust OOD signal from the entire computational path of a VAE. We operationalize this principle by reinterpreting VAEs through the lens of *fast and slow weights*, enabling online, instance-wise inference without costly retraining. Our method extracts interpretable features from the parameters of the VAE's encoder and decoder components—read off the inference path *before* they collapse into the scalar likelihood—and feeds them into a classical density estimator. On standard benchmarks (CIFAR-10, SVHN, CIFAR-100), our LPath method is competitive across the board and, on the hardest pairs, matches or exceeds models with over 10x the parameters at single-forward-pass cost. Our lightweight 3M-parameter VAE provides an efficient and principled approach to real-world, streaming OOD detection.

## 1 Introduction

The assumption that training and test data are independent and identically distributed (IID) underpins much of machine learning's success. However, in real-world deployments, models inevitably encounter out-of-distribution (OOD) data, which can compromise their reliability and safety (Hendrycks & Gimpel, 2016). This challenge is most acute in unsupervised, streaming settings, where labeled data is absent and decisions must be made on-the-fly for each incoming sample. The goal of unsupervised OOD detection is to learn a scoring function from in-distribution (ID) data that can reliably identify anomalous inputs at test time.

A theoretically elegant approach is to use the likelihood $p_\theta(\mathbf{x})$ assigned by a deep generative model (DGM). Intuitively, ID samples should have high likelihood, while OOD samples should have low likelihood. Yet, this intuition famously fails in practice: high-capacity DGMs frequently assign higher likelihoods to OOD samples than to ID samples (Nalisnick et al., 2018; Choi et al., 2018). This paradox has been explored extensively, with studies showing that even perfect density models can be poor OOD indicators (Le Lan & Dinh, 2021; Zhang et al., 2021) and that perfect estimation is often infeasible for modern DGMs (Behrmann et al., 2021; Dai & Wipf, 2019). This leads to our first research question:

> **Research Question 1:** Why does the likelihood score, a cornerstone of statistical modeling, systematically fail for OOD detection in high-capacity deep generative models?

We posit that this failure is particularly revealing in latent variable models like Variational Autoencoders (VAEs). Here, we identify a key culprit: the **Semantic-Texture Likelihood Cancellation Trap**. In a VAE, the final likelihood score is a marginalization of the product of components. For an OOD sample, the decoder might find the low-level texture easy to reconstruct (high conditional likelihood), while the encoder

maps it to a semantically meaningless, low-probability latent code. These opposing signals can *cancel each other out*, yielding a benign final likelihood that masks the anomaly. This insight motivates our second research question:

> **Research Question 2:** Can we extract a more robust OOD signal from the *computational path* of a generative model, rather than from its final scalar output?

To answer this, we propose the **Likelihood Path (LPath) Principle**. It asserts that the intermediate statistics forming the computational path to the likelihood contain more robust information than the final score. This has a classical root, developed in Section 4: where standard likelihood-based scoring keeps only the final number—the *likelihood principle*—we instead keep the component-level quantities that *determine* it, in the spirit of the *sufficiency principle*. We operationalize this principle by reading off the *parameters of a VAE's component distributions*—the encoder and decoder means and variances—and summarizing them with simple norms. These parameters are the natural quantities on the computational path to the likelihood; the norms we extract are deliberately lossy, interpretable features rather than sufficient statistics, and we claim no optimality for them. Being low-dimensional, they remain robust to the curse of dimensionality. By capturing this information *before* the components are multiplied into a single likelihood score, our approach sidesteps the cancellation trap.

Our light-weight inference approach is particularly suitable for streaming settings, since these instance-wise component parameters can be reinterpreted through the lens of *fast and slow weights* (Hinton & Plaut, 1987; Ba et al., 2016). The model's trained parameters $(\theta, \phi)$ are the *slow weights*, learned offline. For each test sample $\mathbf{x}$, the VAE's encoder computes instance-dependent posterior parameters, which we view as *fast weights*. This enables efficient online inference without the retraining required by methods like Likelihood Regret (Xiao et al., 2020).

Our LPath algorithm is a two-stage process. We first use the VAE for fast neural feature extraction of these component parameters, and then feed them into a classical density estimator (e.g., COPOD (Li et al., 2020)) to generate the final OOD score.

This approach achieves competitive and, on the hardest pairs, state-of-the-art OOD detection performance on CIFAR-10, CIFAR-100, and SVHN benchmarks. Crucially, we achieve these results using a lightweight DC-VAE architecture (Xiao et al., 2020) with only **3M** parameters, a fraction of the **44M** for Glow-based DoSE (Morningstar et al., 2021) or **46M** for diffusion models (Liu et al., 2023b). This efficiency makes our method well suited to low-latency, streaming applications.

Our contributions are threefold:

1. **Methodological:** We propose the **LPath Principle**, a novel framework for OOD detection that generalizes the classical likelihood principle by extracting interpretable features from the parameters of the model's likelihood components to overcome likelihood cancellation.

2. **Practical:** We introduce a highly efficient OOD detection algorithm based on a fast-and-slow-weight reinterpretation of VAEs, enabling strong, competitive performance in streaming settings with a lightweight (3M parameter) model.

3. **Empirical:** We demonstrate strong, often state-of-the-art performance on challenging OOD detection benchmarks, validating that our principled, lightweight approach is both effective and efficient.

## 2 Related Work

Our work is situated at the junction of likelihood-based OOD detection, its documented failures, and the subsequent paradigm shift towards probing internal model computations for more robust signals.

**Likelihood-Based OOD Detection.** The use of likelihood scores from DGMs like normalizing flows (Kingma & Dhariwal, 2018) and VAEs (Kingma & Welling, 2013) is a natural starting point. However, the seminal work of Nalisnick et al. (2018) demonstrated their surprising failure. Subsequent works have shown that even with perfect density models, likelihood may be a poor OOD indicator due to geometric factors and

mismatched support (Le Lan & Dinh, 2021; Zhang et al., 2021), and that perfect estimation itself is often infeasible (Behrmann et al., 2021). Likelihood Regret (Xiao et al., 2020) attempted to salvage the likelihood by fine-tuning on test samples and measuring the change in likelihood, but this is computationally prohibitive for streaming applications. Our method avoids this by leveraging "fast weights."

**Probing Internal Model Computations for OOD Signals.** The failure of the marginal likelihood score catalyzed a shift in focus from a model's final output to its internal computational process. This has led to a diverse set of methods that extract OOD signals from deeper within the network. One successful branch of research sources signals from pre-trained *discriminative* models, such as using the Mahalanobis distance in a classifier's feature space (Lee et al., 2018). Another branch, more closely related to our work, probes the internal workings of *generative* models. This includes re-interpreting model outputs as unnormalized energy scores (Liu et al., 2020) or extracting various statistics from the DGM's computational graph, such as Jacobian norms or entropy terms (Morningstar et al., 2021; Graham et al., 2023). While effective, the selection of these statistics is often heuristic. The LPath principle provides a *principled* foundation for this process, grounding the choice of statistics in the classical principles of likelihood and sufficiency.

**Theoretical Foundations.** Our work is grounded in classical statistical theory, specifically the Likelihood Principle (Berger & Wolpert, 1988) and the Sufficiency Principle (Wasserman, 2006; Cvitkovic & Koliander, 2019). We extend these ideas to the context of imperfect deep generative models. Our reinterpretation of VAEs also draws inspiration from the concept of **fast and slow weights**, which has roots in meta-learning (Hinton & Plaut, 1987; Ba et al., 2016) and has recently been connected to attention mechanisms in Transformers (Schlag et al., 2021; Ha et al., 2022).

## 3 The Likelihood Path Method

Our method is built on a reinterpretation of VAEs through the lens of fast and slow weights, guided by our LPath Principle. To build the foundation for our approach, we first clarify the crucial distinction between using a model for static evaluation versus online inference, as this dichotomy is central to understanding both the limitations of prior work and the advantages of our method.

### 3.1 A Tale of Two Inferences: Evaluation vs. Online Adaptation

In the context of OOD detection, a trained deep generative model can be used in two fundamentally different ways at test time.

**Evaluation Procedure (Static Scoring).** The most common approach is a static *evaluation procedure*. Given a new sample $\mathbf{x}$ and a model with pre-trained parameters $\psi_{\text{trained}}$, we compute a scalar score, typically the log-likelihood:

$$(\mathbf{x}, \psi_{\text{trained}}) \longrightarrow p_{\psi_{\text{trained}}}(\mathbf{x}) \in \mathbb{R}. \tag{1}$$

Here, the model parameters $\psi_{\text{trained}}$ are fixed. As discussed, this approach often fails because the scalar score can be misleading due to the likelihood cancellation trap.

**Inferential Procedure (Online Adaptation).** A more sophisticated approach is to perform an *inferential procedure* for each new test sample. In statistics, inference typically refers to estimating model parameters from data. In the context of OOD detection, this means updating the model parameters based on a new test sample $\mathbf{x}_{\text{test}}$:

$$(\mathbf{x}_{\text{test}}, \psi_{\text{trained}}) \longrightarrow \psi_{\text{online}} \in \Psi. \tag{2}$$

The likelihood differences resulted from changes in parameters, $|p_{\psi_{\text{online}}} - p_{\psi_{\text{trained}}}|$, can serve as an OOD score. This is precisely the strategy employed by Likelihood Regret (Xiao et al., 2020), where $\psi_{\text{trained}}$ represents the VAE's network weights (the *slow weights*). While principled, this approach is computationally prohibitive for streaming applications, as it requires retraining by backpropagation for every single sample.

Our work carves out a third path. We perform an online inferential procedure not on the slow network weights, but on a set of instance-dependent *fast weights*. As we will show, VAEs are particularly suited for

this, allowing us to gain the benefits of an inferential approach without the crippling computational cost of retraining.

## 3.2 VAEs as a Vehicle for Fast-Weight Inference

Variational Autoencoders (Kingma & Welling, 2013) are latent variable models that provide the ideal structure for our method. A VAE is defined by a decoder $p_\theta(\mathbf{x}|\mathbf{z})$ and an encoder $q_\phi(\mathbf{z}|\mathbf{x})$. The marginal likelihood $p_\theta(\mathbf{x})$ is approximated using the encoder as an importance sampler:

$$\log p_\theta(\mathbf{x}) \approx \frac{1}{N} \sum_{i=1}^{N} [\log p_\theta(\mathbf{x}|\mathbf{z}_i) + \log p(\mathbf{z}_i) - \log q_\phi(\mathbf{z}_i|\mathbf{x})], \quad \text{where} \quad \mathbf{z}_i \sim q_\phi(\mathbf{z}|\mathbf{x}). \tag{3}$$

This decomposition is the source of the likelihood cancellation trap. The reconstruction term $(\log p_\theta(\mathbf{x}|\mathbf{z}))$ and the regularization terms $(\log p(\mathbf{z}) - \log q_\phi(\mathbf{z}|\mathbf{x}))$ can contain conflicting OOD signals that are neutralized when summed. Our goal is to intercept the information from these components *before* they are combined.

**Gaussian VAEs and Their Component Parameters.** A key to our approach lies in using Gaussian VAEs. Here, the prior, encoder, and decoder are all Gaussian distributions:

$$p(\mathbf{z}) = \mathcal{N}(\mathbf{z} \mid \mathbf{0}, \mathbf{I}), \tag{4}$$

$$q_\phi(\mathbf{z}|\mathbf{x}) = \mathcal{N}(\mathbf{z} \mid \boldsymbol{\mu}_z(\mathbf{x}; \phi), \mathrm{diag}(\boldsymbol{\sigma}_z^2(\mathbf{x}; \phi))), \tag{5}$$

$$p_\theta(\mathbf{x}|\mathbf{z}) = \mathcal{N}(\mathbf{x} \mid \boldsymbol{\mu}_x(\mathbf{z}; \theta), \mathrm{diag}(\boldsymbol{\sigma}_x^2(\mathbf{z}; \theta))). \tag{6}$$

The parameters of these distributions—the means and variances—are not arbitrary activations: they are the *parameters of the component distributions* the VAE emits for each sample, and they are the quantities the likelihood is computed from. Two properties make them an attractive starting point for OOD features:

- **Complete for the components:** together they fully determine the encoder and decoder distributions for a given sample. (They are component-distribution parameters, not sufficient statistics of a data sample in the classical Fisher–Neyman sense.)

- **Compact:** they are far lower-dimensional than the raw activations, which keeps the second stage robust to the curse of dimensionality. The implemented detector reduces them further to simple norms, which are deliberately lossy summaries rather than sufficient statistics.

**Reinterpreting VAE Inference: Fast and Slow Weights.** We can now connect this back to our fast-and-slow-weight framework. The network weights $\phi$ and $\theta$ are the **slow weights**, learned offline. For any new sample $\mathbf{x}_{\text{test}}$, the encoder network performs a rapid, feed-forward pass to compute the parameters of the posterior:

$$(\mathbf{x}_{\text{test}}, \phi_{\text{trained}}) \longrightarrow (\boldsymbol{\mu}_z(\mathbf{x}_{\text{test}}), \boldsymbol{\sigma}_z(\mathbf{x}_{\text{test}}), \boldsymbol{\mu}_x(\mathbf{z}_{\text{test}}), \boldsymbol{\sigma}_x(\mathbf{z}_{\text{test}})). \tag{7}$$

These instance-dependent parameters are the **fast weights** we seek. They are the output of a cheap online *inferential* procedure. Because they are the component parameters read off *before* the scalar aggregation, they are natural candidates for an OOD signal that is both informative and compact, directly addressing the likelihood cancellation trap.

## 3.3 OOD Detection with VAE Fast Weights

Having established that VAEs provide a mechanism for efficient online inference via fast weights—the parameters of the model's component distributions—we now detail how we leverage this for OOD detection. Our goal is to design a scoring function that is effective (by intercepting information before it is lost to the cancellation trap), principled (by reading off the likelihood's component parameters), and efficient (by relying only on fast weights).

### 3.3.1 From Likelihood Components to Their Parameters

Recall the finite sample approximation of the VAE log-likelihood from Equation (3):

$$\log p_\theta(\mathbf{x}) \approx \frac{1}{N} \sum_{i=1}^{N} [\underbrace{\log p_\theta(\mathbf{x}|\mathbf{z}_i)}_{\text{Decoder Term}} + \underbrace{\log p(\mathbf{z}_i)}_{\text{Prior Term}} - \underbrace{\log q_\phi(\mathbf{z}_i|\mathbf{x})}_{\text{Encoder Term}}]$$

The primary likelihood cancellation trap occurs when these three terms are summed. However, a more subtle cancellation can occur even *within* each term. For a multivariate Gaussian distribution with mean $\boldsymbol{\mu}$ and covariance $\boldsymbol{\Sigma}$, the log-likelihood of a sample $\mathbf{y}$ is:

$$\log \mathcal{N}(\mathbf{y} \mid \boldsymbol{\mu}, \boldsymbol{\Sigma}) = -\frac{1}{2} \left( (\mathbf{y} - \boldsymbol{\mu})^\top \boldsymbol{\Sigma}^{-1} (\mathbf{y} - \boldsymbol{\mu}) + \log |\boldsymbol{\Sigma}| + k \log(2\pi) \right). \tag{8}$$

Consider the Mahalanobis distance term, $(\mathbf{y} - \boldsymbol{\mu})^\top \boldsymbol{\Sigma}^{-1} (\mathbf{y} - \boldsymbol{\mu})$. This single scalar value conflates two distinct sources of information: the magnitude of the residual error $(\mathbf{y} - \boldsymbol{\mu})$ and the model's uncertainty about that residual, encoded in $\boldsymbol{\Sigma}$. An OOD sample might produce a large residual but also high uncertainty, while an ID sample might have a small residual and low uncertainty. The final Mahalanobis distance could be similar in both cases, masking the underlying difference. Information about the absolute scale of the error and the uncertainty is lost. This information loss can be summarized as "cancellation behind cancellation".

Therefore, to create a maximally informative OOD score, we propose to bypass the computation of the log-likelihood value entirely. Instead of using the scalar outputs of the likelihood components, we directly use their underlying parameters $(\boldsymbol{\mu}, \boldsymbol{\Sigma})$ that are computed as fast weights.

### 3.3.2 The LPath Statistics: Principled Features from Likelihood Components

The full set of fast weights—the vectors $(\boldsymbol{\mu}_z(\mathbf{x}), \boldsymbol{\sigma}_z(\mathbf{x}))$ from the encoder and $(\boldsymbol{\mu}_x(\mathbf{z}), \boldsymbol{\sigma}_x(\mathbf{z}))$ from the decoder—are the parameters of the VAE's component distributions, but they remain high-dimensional. A direct approach might be to use them directly, but this would be computationally intensive and susceptible to the curse of dimensionality in the second stage of our algorithm (Maciejewski et al., 2022).

Our key insight is that we do not need to summarize these vectors in an ad-hoc way. Instead, we can extract their fundamental properties that *directly determine* the value of the component likelihoods. Recall the log-likelihood of a multivariate Gaussian from Equation (8):

$$\log \mathcal{N}(\mathbf{y} \mid \boldsymbol{\mu}, \boldsymbol{\Sigma}) = -\frac{1}{2} \left( \underbrace{(\mathbf{y} - \boldsymbol{\mu})^\top \boldsymbol{\Sigma}^{-1} (\mathbf{y} - \boldsymbol{\mu})}_{\text{Mahalanobis Distance}} + \underbrace{\log |\boldsymbol{\Sigma}|}_{\text{Log-Determinant}} + k \log(2\pi) \right).$$

The two data-dependent terms are determined by the residual vector $(\mathbf{y} - \boldsymbol{\mu})$ and the uncertainty vector(s) in $\boldsymbol{\Sigma}$. For a *general* diagonal covariance, these terms depend on $\sum_i r_i^2/\sigma_i^2$ and $\sum_i \log \sigma_i^2$ and are *not* determined by the L2 norms alone. Under the isotropic prior and the fixed scalar decoder variance used in our implementation (Appendix D.1), however, the reconstruction term is an exact function of $u = \|\mathbf{x} - \hat{\mathbf{x}}\|_2$ and the prior term an exact function of $(v, w)$; the only ingredient an L2 norm does not capture is the encoder entropy $\frac{1}{2} \sum_i \log \sigma_{z,i}^2$. We therefore treat these norms as interpretable *features* motivated by the likelihood decomposition—capturing the dominant "ingredients" of the component likelihoods *before* they are combined—rather than as a lossless reduction; Appendix D.3 introduces $\ell^p/\ell^q$ norms to recover some of the coordinate-wise information the L2 norm discards.

This choice leads to our LPath statistics. We apply this logic to both the encoder and decoder components of the VAE, yielding a practical, low-dimensional feature vector. In principle the decoder also contributes a reconstruction-standard-deviation norm $s(\mathbf{x})$; however, because we use a fixed scalar decoder variance (Appendix D.1), $s(\mathbf{x})$ is constant across inputs and is dropped, leaving the three-dimensional $T(\mathbf{x}) =$

---

**Algorithm 1** The LPath Algorithm for OOD Detection

---

1: **Input:** Training data $\mathcal{D}_{\text{ID}}$, test data $\mathcal{D}_{\text{Test}}$, classical OOD algorithm $\mathcal{A}_{\text{classical}}$.

2: **Stage 1: Neural Feature Extraction (Offline Training)**
3: Train VAE(s) on $\mathcal{D}_{\text{ID}}$ using a standard VAE loss.
4: Initialize feature set $\mathcal{T}_{\text{ID}} \leftarrow \emptyset$.
5: **for** each sample $\mathbf{x} \in \mathcal{D}_{\text{ID}}$ **do**
6:     Compute LPath statistics $T(\mathbf{x}) = (u(\mathbf{x}), v(\mathbf{x}), w(\mathbf{x}))$ using Equations (9) to (11).
7:     Add the 3-dim vector $T(\mathbf{x})$ to $\mathcal{T}_{\text{ID}}$.
8: **end for**

9: **Stage 2: Classical Density Estimation (Offline Training)**
10: Train $\mathcal{A}_{\text{classical}}$ on the low-dimensional feature set $\mathcal{T}_{\text{ID}}$ to get a trained detector $\mathcal{A}_{\text{trained}}$. ▷ e.g., Estimate copulas for COPOD or mean/covariance for MD.

11: **Inference (Online and Fast)**
12: **for** each sample $\mathbf{x}_{\text{test}} \in \mathcal{D}_{\text{Test}}$ **do**
13:     Compute statistics $T_{\text{test}} = (u(\mathbf{x}_{\text{test}}), v(\mathbf{x}_{\text{test}}), w(\mathbf{x}_{\text{test}}))$ via a single forward pass.
14:     Compute OOD score $S(\mathbf{x}_{\text{test}}) = \mathcal{A}_{\text{trained}}(T_{\text{test}})$.
15: **end for**
16: **Output:** OOD scores $S$ for all samples in $\mathcal{D}_{\text{Test}}$.

---

$[u(\mathbf{x}), v(\mathbf{x}), w(\mathbf{x})]^T$, where:

$$u(\mathbf{x}) = \|\mathbf{x} - \boldsymbol{\mu}_x(\boldsymbol{\mu}_z(\mathbf{x}))\|_2 \qquad \text{(Reconstruction Error Norm)} \qquad (9)$$

$$v(\mathbf{x}) = \|\boldsymbol{\mu}_z(\mathbf{x})\|_2 \qquad \text{(Latent Mean Norm)} \qquad (10)$$

$$w(\mathbf{x}) = \|\boldsymbol{\sigma}_z(\mathbf{x})\|_2 \qquad \text{(Latent Std Dev Norm)} \qquad (11)$$

Each statistic is not only a building block of a likelihood term but is also directly interpretable in the context of the VAE's training objective. For an in-distribution (ID) sample, we expect:

- A small reconstruction error norm $u(\mathbf{x})$, as the model should reconstruct ID data well. We use the error norm rather than the norm of the reconstruction itself, $\|\boldsymbol{\mu}_x(\dots)\|_2$, to create a naturally centered statistic.

- A small latent mean norm $v(\mathbf{x})$, as the KL-divergence term in the VAE loss regularizes the posterior mean towards the prior mean of zero.

- A latent standard deviation norm $w(\mathbf{x})$ close to a constant determined by the latent dimension, as the KL loss regularizes the posterior variance towards the prior's identity covariance.

Deviations from these expected values provide a strong, multi-faceted signal for OOD detection. This three-dimensional feature vector, derived directly from the parameters of the likelihood components, forms the core of our LPath algorithm.

### 3.3.3 The Two-Stage LPath Algorithm

Our method combines deep representation learning with classical statistics in a two-stage process, as detailed in Algorithm 1. In the first stage, we use a trained VAE for neural feature extraction, computing the three-dimensional LPath statistics for each data point. In the second stage, we use these low-dimensional features to train a classical, non-parametric OOD detection algorithm like COPOD (Li et al., 2020) or a simple Mahalanobis distance (MD) detector.

We propose two practical variants of this algorithm:

- **LPath-1M:** Uses a single VAE to extract all three statistics. This is the simplest and most efficient variant.

- **LPath-2M:** Uses two VAEs to resolve the inherent trade-off between encoder expressivity (which prefers a high-dimensional latent space) and decoder sensitivity (which prefers a low-dimensional one). We use a high-dimensional VAE for encoder statistics $(v, w)$ and a low-dimensional VAE for decoder statistics $(u, s)$. A detailed rationale is provided in Appendix B.

This two-stage design effectively leverages the strengths of both deep learning (for powerful, automatic feature extraction) and classical statistics (for robust, low-dimensional density estimation). In the next section, we will formalize the theoretical underpinnings of this approach with the Likelihood Path Principle.

## 4 The Likelihood Path Principle

Our method instantiates a single idea: the computational *path* to the likelihood carries more information than the scalar score it collapses to, which may be useful for OOD detection. We now make that idea explicit as the **Likelihood Path (LPath) Principle** by grounding it in the classical likelihood and sufficiency principles. The principle is what the method's feature choice follows from—it locates *where* in the model's computation the OOD signal lives, and why collapsing to the scalar likelihood obscures it—and we import these classical ideas to the setting of modern, high-capacity, and imperfectly trained deep generative models.

### 4.1 Foundations: The Classical Likelihood and Sufficiency Principles

To understand our contribution, we first revisit two cornerstones of classical statistical inference.

**The Likelihood Principle.** A central tenet of statistical inference is the *Likelihood Principle* (Berger & Wolpert, 1988). It states that all the evidence in an observed sample $\mathbf{x}$ relevant to the model parameters $\psi$ is contained entirely within the likelihood function, $\ell(\psi|\mathbf{x}) \equiv p(\mathbf{x}|\psi)$. The standard practice of training generative models via Maximum Likelihood Estimation (MLE) directly adheres to this principle, as the choice of optimal parameters $\psi_{\mathrm{MLE}}$ depends only on this function. Many likelihood-based OOD detection methods (Nalisnick et al., 2018; Xiao et al., 2020) implicitly follow this principle by using the final likelihood score, which is a functional of $\ell(\psi|\mathbf{x})$, as their OOD signal.

**The Sufficiency Principle.** While the likelihood function contains all the information, it is often a complex, high-dimensional object. The *Sufficiency Principle* provides a formal way to compress this information without loss. A statistic $T(\mathbf{x})$ is called a *sufficient statistic* for $\psi$ if the conditional distribution of the data given the statistic, $p(\mathbf{x}|T(\mathbf{x}), \psi)$, does not depend on $\psi$. In other words, once $T(\mathbf{x})$ known, the original data $\mathbf{x}$ provides no further information about the parameters $\psi$. A sufficient statistic is *minimal* if it is the most compressed representation possible without losing information (Cvitkovic & Koliander, 2019). Minimal sufficient statistics provide the ideal trade-off: they are maximally informative yet minimally complex, helping to avoid the curse of dimensionality that plagues methods using less-principled, high-dimensional features.

### 4.2 An Information-Theoretic View of the Likelihood Path

These classical principles provide a powerful lens. Figure 1 illustrates this classical view, showing a sequence of steps where information about the data is progressively compressed, from the raw data $\mathbf{x}$ to the likelihood function and finally to a sufficient statistic $T(\mathbf{x})$. To formalize this notion of information reduction and apply it to our VAE context, we turn to the information theoretic perspective of sufficient statistics. The relationship between data, statistics, and parameters can be quantified using mutual information, $\mathrm{I}(\cdot, \cdot)$. A statistic $T(\mathbf{x})$ is sufficient for parameters $\psi$ if and only if it preserves all the mutual information between the data and the parameters: $\mathrm{I}(\psi; T(\mathbf{x})) = \mathrm{I}(\psi; \mathbf{x})$. This lens is more adaptable and inspires our LPath principle.

The abstract information pathway in Figure 1 finds a concrete analogue in the computational process of a VAE. Recall the VAE likelihood calculation:

$$p_\theta(\mathbf{x}) = \int_{\mathbf{z} \sim P(\mathbf{z})} p_\theta(\mathbf{x} \mid \mathbf{z}) p(\mathbf{z}) \, \mathrm{d}\mathbf{z} = \int_{\mathbf{z} \sim q_\phi(\mathbf{z}|\mathbf{x})} \frac{p_\theta(\mathbf{x} \mid \mathbf{z}) p(\mathbf{z})}{q_\phi(\mathbf{z} \mid \mathbf{x})} \, \mathrm{d}\mathbf{z}, \tag{12}$$

For any given sample $\mathbf{x}$ and any latent variable $\mathbf{z}$, the VAE likelihood computation process can be viewed as an integral of Markov chain of information processing:

$$\mathbf{x} \longrightarrow \underbrace{\{(\boldsymbol{\mu}_z(\mathbf{x}), \boldsymbol{\sigma}_z(\mathbf{x})), (\boldsymbol{\mu}_x(\mathbf{z}), \boldsymbol{\sigma}_x(\mathbf{z}))\}}_{\text{Component Parameters (Fast Weights)}} \longrightarrow \underbrace{\{q_\phi(\mathbf{z}|\mathbf{x}), p_\theta(\mathbf{x}|\mathbf{z}), p(\mathbf{z})\}}_{\text{Component Distributions}} \longrightarrow \underbrace{p_\theta(\mathbf{x})}_{\text{Final Likelihood Score}} \tag{13}$$

where $\mathbf{z} \sim q_\phi(\mathbf{z}|\mathbf{x})$. The above naturally admits the following sampling version, where we sample $N$ latent codes:

$$\mathbf{x} \longrightarrow \underbrace{\{(\boldsymbol{\mu}_z(\mathbf{x}), \boldsymbol{\sigma}_z(\mathbf{x})), (\boldsymbol{\mu}_x(\mathbf{z}_i), \boldsymbol{\sigma}_x(\mathbf{z}_i))\}_{i=1}^N}_{\text{Sampling Component Parameters}} \longrightarrow \underbrace{\{q_\phi(\mathbf{z}_i|\mathbf{x}), p_\theta(\mathbf{x}|\mathbf{z}_i), p(\mathbf{z}_i)\}_{i=1}^N}_{\text{Component Sampling Distributions}} \longrightarrow \underbrace{p_\theta(\mathbf{x})}_{\text{Estimated Likelihood}} \tag{14}$$

The sampling version is rigorously a Markov chain of information processing. Let's denote these component parameters as $\mathcal{T}(\mathbf{x}) = \{(\boldsymbol{\mu}_z(\mathbf{x}), \boldsymbol{\sigma}_z(\mathbf{x})), (\boldsymbol{\mu}_x(\mathbf{z}), \boldsymbol{\sigma}_x(\mathbf{z}))\}$ and the set of component sampling distributions as $\mathcal{P}(\mathbf{x})$. By the Data Processing Inequality, any processing step cannot increase information. This gives us a chain of inequalities that mirrors the information reduction shown in Figure 1:

$$\mathrm{I}(\mathbf{x}; \mathcal{T}(\mathbf{x})) \geq \mathrm{I}(\mathbf{x}; \mathcal{P}(\mathbf{x})) \geq \mathrm{I}(\mathbf{x}; p_\theta(\mathbf{x})). \tag{15}$$

Let's analyze these inequalities.

1. $\mathrm{I}(\mathbf{x}; \mathcal{T}(\mathbf{x})) \geq \mathrm{I}(\mathbf{x}; \mathcal{P}(\mathbf{x}))$: Since the component distributions $\mathcal{P}(\mathbf{x})$ are completely determined by the component parameters $\mathcal{T}(\mathbf{x})$, no information is lost in this step. Therefore, $\mathrm{I}(\mathbf{x}; \mathcal{T}(\mathbf{x})) = \mathrm{I}(\mathbf{x}; \mathcal{P}(\mathbf{x}))$. The fast weights contain exactly as much information as the full distributions they parameterize.

2. $\mathrm{I}(\mathbf{x}; \mathcal{P}(\mathbf{x})) \geq \mathrm{I}(\mathbf{x}; p_\theta(\mathbf{x}))$: This is the crucial step. The final likelihood $p_\theta(\mathbf{x})$ is computed by integrating/multiplying the component distributions in $\mathcal{P}(\mathbf{x})$. If the model were a perfect representation of the data generating process, this step might also preserve information. However, for a real-world, imperfectly trained DGM, this is where the **likelihood cancellation** occurs. As opposing signals from the encoder and decoder are combined, information about the anomaly is irreversibly destroyed.

This motivates the central *intuition* of our work—a necessary condition, not a guarantee. Under imperfect density estimation the second inequality is generically strict,

$$\mathrm{I}(\mathbf{x}; \mathcal{P}(\mathbf{x})) > \mathrm{I}(\mathbf{x}; p_\theta(\mathbf{x})). \tag{16}$$

The core of our method is to exploit this information gap. We stress what this does *not* establish: the inequality concerns information about $\mathbf{x}$ in general, and that the preserved information is *useful for distinguishing ID from OOD* is not implied by the data-processing inequality—it is an empirical claim, which we establish in Section 5.3 (Table 2) and Section 5.2 (Table 1). By extracting features from the components *before* they are combined, we access a richer source of information whose OOD-relevance we then verify empirically.

### 4.3 The Analogy: Reading the Path, Not the Score

The two classical principles and the VAE information chain now line up, and the analogy between them is the heart of our framework. The *likelihood principle* says all evidence about the parameters lives in the likelihood; the *sufficiency principle* says we may compress the data to whatever *determines* that likelihood, losing nothing. Read together, they license a two-move recipe: form the likelihood, then keep only what determines it.

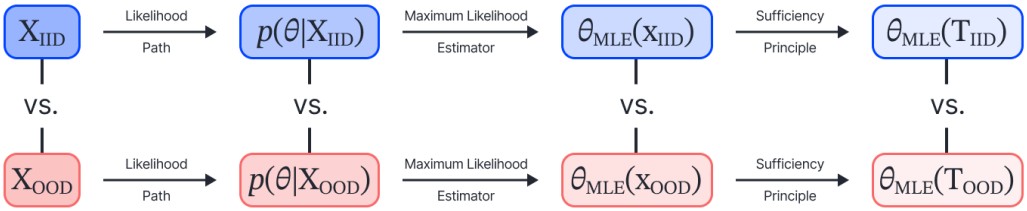

Figure 1: A conceptual diagram of the classical statistical inference pipeline, contrasting the paths for in-distribution ($\mathbf{x}_{\text{ID}}$, top) and out-of-distribution ($\mathbf{x}_{\text{OOD}}$, bottom) data. The diagram illustrates the progressive information reduction from left to right by applying the Likelihood Principle (LP), Maximum Likelihood Estimation (MLE), and the Sufficiency Principle (SP). $T$ denotes the sufficient statistic. The LPath method is inspired by analyzing the differences between these two paths.

Likelihood-based OOD detection scoring inherits the first move. While the likelihood principle studies the likelihood function as a function of the parameters, likelihood OOD scoring treats the likelihood function as a function of the data $\mathbf{x}$. Treating the scalar $p_\theta(\mathbf{x})$ as the complete evidence about whether $\mathbf{x}$ is anomalous is reminiscent of applying the likelihood function without considering what determines it. This is where the analogy strains: for an imperfect VAE the collapse to a scalar is lossy (the cancellation of Equation (16), made concrete in Section 4.5), so the scalar is no longer a faithful summary of the evidence the way it is in the classical, well-specified world.

The second move is what we adopt. The quantities that *determine* the VAE's likelihood computation are not the scalar but the per-sample component parameters $\mathcal{T}(\mathbf{x})$ —the encoder and decoder Gaussian means and variances from which every factor of $p_\theta(\mathbf{x})$ is built. Tracking these, rather than the number they are eventually aggregated into, is the sufficiency-inspired move: keep what determines the likelihood, *before* aggregation discards the cross-component structure. The likelihood path $\mathbf{x} \rightarrow \mathcal{T}(\mathbf{x}) \rightarrow \mathcal{P}(\mathbf{x}) \rightarrow p_\theta(\mathbf{x})$ (Equation (13)) is the VAE analogue of the classical compression pipeline $\mathbf{x} \rightarrow \ell(\psi \mid \mathbf{x}) \rightarrow T(\mathbf{x})$ (Figure 1), with one decisive difference: classically the compression to $T$ is *lossless* for the likelihood, whereas in the VAE the lossy step is the collapse to the *scalar*—so we stop one stage earlier, at the component parameters. Our premise is that this sufficiency principle analogue is more adaptable in the OOD setting.

Two caveats keep the analogy honest. First, it is an analogy, not an instance of classical sufficiency: $\mathcal{T}(\mathbf{x})$ are sample-dependent outputs of a fixed network, not Fisher–Neyman sufficient statistics for an unknown population parameter, and the information they preserve is about $\mathbf{x}$ for this fixed model, not about $\psi$—which is why the chain in Equation (15) is written in $\mathrm{I}(\mathbf{x}; \cdot)$ rather than $\mathrm{I}(\psi; \cdot)$. Second, the full parameters $\mathcal{T}(\mathbf{x})$ are what this path-level argument concerns; the features our detector actually implements are simple norms $(u, v, w)$ (Section 3.3.2), which are deliberately *lossy* summaries—not sufficient, not even path-sufficient—and for which we claim no optimality.

## 4.4 The LPath Principle as a Generalization

The analogy of Section 4.3 motivates our principle, and the information chain of Section 4.2 formalizes it. We can now state it.

**Definition 4.1** (The Likelihood Path (LPath) Principle)**.** *Principle (general).* For a latent-variable model under imperfect density estimation, the parameters of the likelihood's factorization components can carry more information about the input than the final scalar marginal likelihood, which is subject to information loss via cancellation. *Instantiation (this paper).* For a Gaussian VAE we read these component parameters off the inference path and summarize them with simple norms; whether the retained information improves OOD detection is an empirical question, which our experiments answer affirmatively. We state the principle as motivation rather than as a theorem, and make no sufficiency or optimality claim for the implemented norms.

The LPath Principle is a natural generalization of the classical Likelihood Principle for the modern era of deep generative models. The likelihood principle assumes a perfect, well-specified model and suggests using the likelihood function. Our principle acknowledges the reality of imperfect models and suggests using the

more informative building blocks of that likelihood function. The path of component parameters, $\mathcal{T}(\mathbf{x})$, is a useful compromise: it is informative about the components of the likelihood, low-dimensional, and avoids the information destruction of the final integration step. This motivates our method, which is based on simple L2 norms of these parameters; the norms themselves are lossy features for which we claim no optimality.

### 4.5 The Mechanism of Likelihood Cancellation

While Section 4 provides a **statistical** view on information loss, a **combinatorial** perspective offers a more concrete mechanism: **arithmetic cancellation**. In a VAE, the final likelihood score is a product of components that can carry opposing signals for OOD detection. An anomalous signal from one component can be neutralized by a benign signal from another, masking the OOD nature of the input.

This phenomenon is rooted in the VAE's approximate log-likelihood, $\log p_\theta(\mathbf{x}) \approx \log p_\theta(\mathbf{x} \mid \mathbf{z}) + \log p(\mathbf{z}) - \log q_\phi(\mathbf{z} \mid \mathbf{x})$. We can interpret this cancellation through the lens of a texture-semantic conflict:

- The **decoder's conditional likelihood**, $p_\theta(\mathbf{x} \mid \mathbf{z})$, often focuses on reconstructing low-level **pixel textures**.
- The **prior evaluated at the latent code**, $p(\mathbf{z})$ for $\mathbf{z} \sim q_\phi(\mathbf{z} \mid \mathbf{x})$, captures higher-level **semantics**.

This is the **Semantic-Texture Likelihood Cancellation Trap** mentioned in Section 1, leading to two common failure modes. First, consider an OOD sample that is semantically different but has familiar textures (e.g., an image of a dog with the texture of cat fur). The decoder may reconstruct it well (high $p_\theta(\mathbf{x} \mid \mathbf{z})$), while the encoder maps it to an unlikely, semantically incorrect latent code (low $p(\mathbf{z})$). As noted by Havtorn et al. (2021), the final likelihood is often dominated by low-level texture information, so the strong reconstruction signal can overwhelm the weak semantic signal. Conversely, consider an OOD sample with unfamiliar textures, leading to a large reconstruction error (low $p_\theta(\mathbf{x} \mid \mathbf{z})$). This provides a strong OOD signal. However, if the sample happens to map to a region of unusually high prior density, this can partially offset the low reconstruction likelihood, making the sample appear less anomalous than it is.

We conceptualize these different interactions as distinct **likelihood paths**, which are illustrated conceptually in Figure 2. This cancellation effect is not merely theoretical; it is directly observable in the data. Figure 3 provides a striking empirical demonstration. The distributions of the individual LPath statistics—the reconstruction error in Figure 3(b) and the latent norm in Figure 3(a)—show a relatively clear separation between the ID (CIFAR-10) and OOD (SVHN) data. However, when these signals are combined into the final ELBO score, shown in Figure 3(c), the separation is visibly diminished. The distinct OOD signals have been diluted.

This observation is quantified in our experimental analysis in Section 5.3, where we map the four conceptual cases from Figure 2 to these component behaviors and show that the ELBO's performance degrades significantly in precisely the scenarios where cancellation is expected (Table 2). The LPath method is designed to circumvent this problem by capturing the rich, multi-faceted information from the component statistics *before* they are collapsed into a single, less informative score.

## 5 Experiments

We conduct a series of experiments to validate the LPath method. First, we compare its OOD detection performance against state-of-the-art methods on standard benchmarks. Second, we perform a targeted analysis to empirically demonstrate the likelihood cancellation phenomenon and show how the LPath statistics overcome it.

### 5.1 Experimental Setup

**Baselines and Setting.** We compare our LPath method with a suite of state-of-the-art OOD detection methods, including those based on likelihood-regret (LR), hierarchical VAEs (BIVA), normalizing flows (DoSE), diffusion models (DDPM), and other feature-based approaches (Xiao et al., 2020; Havtorn et al., 2021;

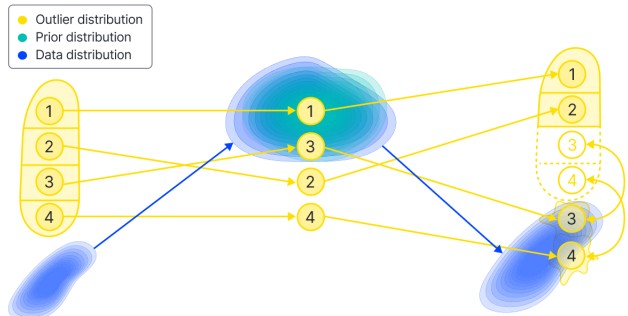

Figure 2: **Conceptual illustration of different likelihood paths.** This figure traces in-distribution ($\mathbf{x}_{\text{ID}}$, blue) and out-of-distribution ($\mathbf{x}_{\text{OOD}}$, yellow) data through a VAE's computational path: from the input space (left), through the latent space (middle), to the reconstruction space (right). We identify four distinct cases for OOD data based on their interaction with the prior (turquoise) and their final reconstruction quality. Cases (1) and (2) are well-reconstructed, while Cases (3) and (4) are poorly reconstructed (visualized as 'fried egg' shapes), revealing a strong OOD signal from the decoder. The grey areas highlight *atypical sets* (Nalisnick et al., 2019)—pathological regions of high density but low volume that can mislead likelihood-based OOD detectors.

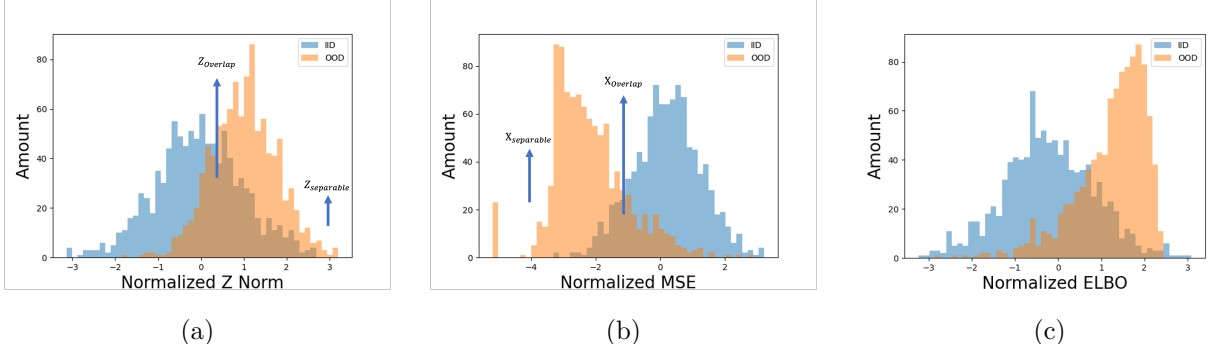

| (a) | (b) | (c) |

Figure 3: Histograms demonstrating the **likelihood cancellation effect** on a VAE with a 100-dimensional latent space, trained on CIFAR-10 (ID) vs. SVHN (OOD). The plots show distributions for three components of the VAE log-likelihood from Equation (3): (a) the latent code norm, $\|\mathbf{z}\|_2$, related to the prior term; (b) the reconstruction error (MSE), $\|\mathbf{x} - \widehat{\mathbf{x}}\|_2$, related to the decoder's conditional likelihood; and (c) the final ELBO score. The four conceptual cases from Figure 2 map to combinations of separability in these components: Case (1) is $Z_{\text{overlap}} + X_{\text{overlap}}$; Case (2) is $Z_{\text{separable}} + X_{\text{overlap}}$; Case (3) is $Z_{\text{overlap}} + X_{\text{separable}}$; and Case (4) is $Z_{\text{separable}} + X_{\text{separable}}$. Note that the separation between ID and OOD data is much clearer in the individual components (a, b) than in the final ELBO (c), illustrating how information is lost. All values are normalized; see Appendix D for details.

Morningstar et al., 2021; Bergamin et al., 2022; Liu et al., 2023b; Graham et al., 2023). All experiments are conducted under a challenging and realistic setting: unsupervised, single-sample, and without any inductive bias about the nature of the OOD data. Crucially, all model selection—including the latent dimension—uses in-distribution validation loss only; no OOD or test data informs any hyperparameter choice (Appendix D).

**Datasets and Metrics.**  Following convention, we use several common image benchmarks. We evaluate performance using CIFAR-10, SVHN, MNIST, and FashionMNIST (FMNIST) as in-distribution (ID) datasets. For OOD data, we use these datasets against each other, as well as CIFAR-100 and synthetically challenging variants like horizontally-flipped (Hflip) and vertically-flipped (Vflip) images. Performance is measured using the Area Under the Receiver Operating Characteristic curve (AUROC). Further details on the VAE architecture and training are in Appendix D.

Figure 4: **Visualization of the data partitioning method for the four-case analysis.** This figure illustrates how we define the 'separable' and 'overlap' regions for the experiment in Section 5.3. The partitioning is based on the modes (dashed vertical lines) of the in-distribution (ID, blue) and out-of-distribution (OOD, orange) data for the reconstruction error ($u(\mathbf{x})$) and latent norm ($v(\mathbf{x})$). **Top Row (a, b):** The original, full distributions of the two statistics. **Middle Row (c, d):** The 'separable' regions, containing data points that fall outside the modes of the opposing distribution. **Bottom Row (e, f):** The 'overlap' regions, containing data points that fall between the two modes. This partitioning creates the four scenarios analyzed in Table 2.

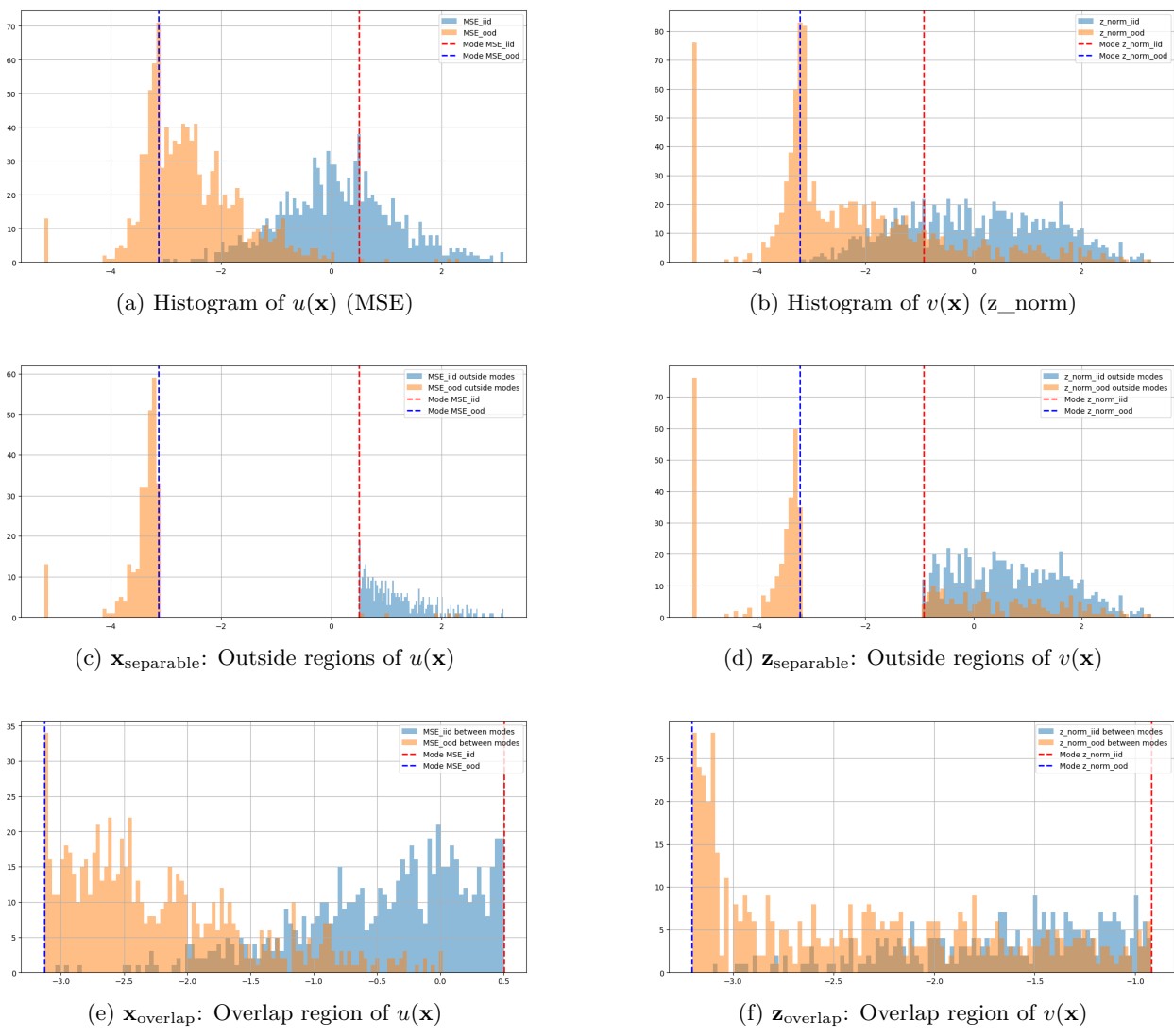

(a) Histogram of $u(\mathbf{x})$ (MSE)

(b) Histogram of $v(\mathbf{x})$ (z_norm)

(c) $\mathbf{x}_{\text{separable}}$: Outside regions of $u(\mathbf{x})$

(d) $\mathbf{z}_{\text{separable}}$: Outside regions of $v(\mathbf{x})$

(e) $\mathbf{x}_{\text{overlap}}$: Overlap region of $u(\mathbf{x})$

(f) $\mathbf{z}_{\text{overlap}}$: Overlap region of $v(\mathbf{x})$

## 5.2 Main Results: State-of-the-Art OOD Detection

**Competitive Performance with a Lightweight Model.** As shown in Table 1, our LPath method (in both its 1-Model and 2-Model variants) is competitive across all benchmark settings. Among unsupervised, single-sample DGM methods it matches or exceeds models with $>10\times$ the parameters on the hardest pairs (e.g., CIFAR-10 vs. SVHN and CIFAR-10 vs. CIFAR-100), at single-forward-pass cost. The method excels on difficult pairs like CIFAR-10 vs. SVHN and demonstrates strong performance on the subtle Hflip/Vflip tasks, where many prior methods perform near chance. This is particularly noteworthy because these flipped datasets differ from the ID data by only one latent dimension, making them a stringent test of a method's sensitivity.

Table 1: **Our LPath method matches or exceeds much larger models on the hardest OOD pairs while using a fraction of the model parameters.** This table presents the Area Under the Receiver Operating Characteristic curve (AUROC) for various methods on common OOD detection benchmarks. The highest score in each column is bolded. Entries marked **N/A** are (ID, OOD) pairs not reported in the cited source paper; we quote authors' published numbers to credit each method with its best reported result. Our methods are presented in three variants: 'LPath-1M' uses a single VAE, while 'LPath-2M' uses a pair of VAEs with different latent dimensions (see Appendix D). 'COPOD' and 'MD' refer to the classical density estimator used in the second stage of our algorithm. Our lightweight VAE-based approach is competitive with, and on the hardest pairs (e.g., CIFAR-10 vs. SVHN) matches or exceeds, much larger models such as DoSE (Glow-based) and DDPM.

| ID
OOD | CIFAR10
SVHN | CIFAR100 | Hflip | Vflip | SVHN
CIFAR10 | Hflip | Vflip | FMNIST
MNIST | Hflip | Vflip | MNIST
FMNIST | Hflip | Vflip |
|---|---|---|---|---|---|---|---|---|---|---|---|---|---|
| ELBO | 0.08 | 0.54 | 0.5 | 0.56 | **0.99** | 0.5 | 0.5 | 0.87 | 0.63 | 0.83 | **1.00** | 0.59 | 0.6 |
| LR (Xiao et al., 2020) | 0.88 | N/A | N/A | N/A | 0.92 | N/A | N/A | 0.99 | N/A | N/A | N/A | N/A | N/A |
| BIVA (Havtorn et al., 2021) | 0.89 | N/A | N/A | N/A | **0.99** | N/A | N/A | 0.98 | N/A | N/A | **1.00** | N/A | N/A |
| DoSE (Morningstar et al., 2021) | 0.97 | 0.57 | 0.51 | 0.53 | **0.99** | 0.52 | 0.51 | **1.00** | 0.66 | 0.75 | **1.00** | **0.81** | 0.83 |
| Fisher (Bergamin et al., 2022) | 0.87 | 0.59 | N/A | N/A | N/A | N/A | N/A | 0.96 | N/A | N/A | N/A | N/A | N/A |
| DDPM (Liu et al., 2023b) | 0.98 | N/A | 0.51 | 0.63 | **0.99** | **0.62** | **0.58** | 0.97 | 0.65 | **0.89** | N/A | N/A | N/A |
| LMD (Graham et al., 2023) | **0.99** | 0.61 | N/A | N/A | 0.91 | N/A | N/A | 0.99 | N/A | N/A | **1.00** | N/A | N/A |
| LPath-1M-COPOD (Ours) | **0.99** | **0.62** | **0.53** | 0.61 | **0.99** | 0.55 | 0.56 | **1.00** | 0.65 | 0.81 | **1.00** | 0.65 | **0.87** |
| LPath-2M-COPOD (Ours) | 0.98 | **0.62** | **0.53** | **0.65** | 0.96 | 0.56 | 0.55 | 0.95 | **0.67** | 0.87 | **1.00** | 0.77 | 0.78 |
| LPath-1M-MD (Ours) | **0.99** | 0.58 | 0.52 | 0.60 | 0.95 | 0.52 | 0.52 | 0.97 | 0.63 | 0.82 | **1.00** | 0.75 | 0.76 |

Table 2: **LPath method excels across all OOD scenarios, especially in the most difficult case.** This table shows the OOD detection performance (AUROC) of different statistics across the four conceptual cases defined in Section 4.5. The cases range from most difficult (Case 1, where ID/OOD distributions overlap in both reconstruction and latent spaces) to easiest (Case 4, where they are separable). The LPath method ("Ours") consistently outperforms the individual component statistics ($u(\mathbf{x})$, $v(\mathbf{x})$) and the final ELBO score, demonstrating its ability to overcome the likelihood cancellation that degrades ELBO's performance in Case 1.

| Statistic Used for OOD Detection | Case 1 | Case 2 | Case 3 | Case 4 |
|---|---|---|---|---|
| Latent Norm ($v(\mathbf{x})$) | 0.75 | 0.74 | 0.93 | 0.96 |
| Reconstruction Error ($u(\mathbf{x})$) | 0.93 | 0.98 | 1.00 | 0.98 |
| ELBO (Final Likelihood Score) | 0.83 | 0.87 | 1.00 | 0.96 |
| **LPath Statistics (Ours)** | **0.99** | **0.99** | **1.00** | **0.99** |

**Efficiency.** This improvement is notable given that we use a very small VAE architecture. Compared to these baselines, our approach relies on a much smaller model (a DC-VAE from Xiao et al. (2020)'s architecture) with a parameter count of only **3M**. This is a fraction of the **44M** parameters required for the Glow model (Kingma & Dhariwal, 2018) used in DoSE (Morningstar et al., 2021) or the **46M** parameters for the diffusion model used in recent work (Rombach et al., 2022; Liu et al., 2023b). Our method clearly exceeds other VAE-based approaches (Xiao et al., 2020; Havtorn et al., 2021) and is the only one that is competitive with these much larger models.

The comparison with DoSE is especially revealing. The authors of DoSE reported results using a VAE for simpler datasets like MNIST/FMNIST but switched to the larger Glow model for more complex datasets like CIFAR-10 and SVHN. This suggests that their heuristically chosen statistics were not sufficiently powerful to achieve SOTA performance on a VAE with challenging data. In contrast, our LPath method, guided by the principled selection of likelihood-component features, surpasses their Glow-based results using our lightweight VAE. While Glow's likelihood estimation is arguably superior to our small VAE's, and DoSE's chosen statistics are sophisticated, our simpler and more principled method is more effective and substantially

cheaper at inference. This showcases that the power of the LPath method lies not in model scale, but in its information-theoretic foundation. This answers RQ2.

**Confidence intervals and significance.** Because per-sample scores from the original runs are unavailable, we report analytic confidence intervals computed from the published AUROCs and the (standard, known) test-set sizes using the Hanley & McNeil (1982) estimator; these capture test-set sampling variability, not training-seed variability (which would require retraining we cannot rerun). With test sets of $n = 10{,}000$–$26{,}032$ the intervals are tight and the headline gaps far exceed sampling noise (Table 3): on CIFAR-10 vs. SVHN the LPath–ELBO gap exceeds 400 standard errors; on CIFAR-10 vs. CIFAR-100 LPath is clearly above the ELBO ($z \approx 14$) but *statistically indistinguishable* from the best baseline LMD ($z \approx 1.8$), which we report as a tie rather than a win; on the flip tasks all methods are near chance and we make no strong claim. The cancellation contrast (Table 2) is likewise robust: even at a conservative $n \approx 500$/class, LPath (0.99) vs. ELBO (0.83) gives $z \approx 12$.

Table 3: Analytic 95% confidence intervals (Hanley–McNeil) for representative CIFAR-10 (ID) comparisons, computed from the reported AUROCs and test-set sizes. Large gaps clear sampling noise by wide margins, while CIFAR-100 vs. the best baseline is a statistical tie.

| Comparison (CIFAR-10 ID) | LPath-1M | 95% CI | ELBO | best baseline | LPath vs. ELBO |
|---|---|---|---|---|---|
| vs. SVHN ($n_{\mathrm{OOD}}$=26,032) | 0.99 | $\pm 0.001$ | 0.08 | 0.98 (DoSE/DDPM) | $z > 400$ |
| vs. CIFAR-100 ($n$=10,000) | 0.62 | $\pm 0.008$ | 0.54 | 0.61 (LMD) | $z \approx 14$ |
| vs. Vflip ($n$=10,000) | 0.61 | $\pm 0.008$ | 0.56 | 0.63 (DDPM) | $z \approx 9$ |

**Inference cost.** Parameter count is not the same as speed, so we frame efficiency structurally rather than as a wall-clock comparison. LPath inference is a single forward pass of a $\sim$3M-parameter VAE followed by an $O(d)$ classical score on a $d$=3 vector. By contrast, Likelihood Regret (Xiao et al., 2020) performs per-sample optimization (repeated forward and backward passes) for every test point; diffusion detectors (Liu et al., 2023b; Graham et al., 2023) require many denoising/inpainting passes over a $\sim$46M-parameter network; and Glow-based DoSE (Morningstar et al., 2021) evaluates many invertible layers of a $\sim$44M-parameter model. These are structural, per-test-sample differences in the dominant computation; we therefore restrict our claim to inference cost and do not assert wall-clock superiority over every baseline.

### 5.3 Analysis: Overcoming Likelihood Cancellation

To empirically validate our central claim that the final likelihood score suffers from information loss, we conduct an experiment based on the four conceptual OOD cases illustrated in Figure 2.

**Experimental Design.** We partition the OOD data (SVHN) into four cases based on whether their LPath statistics fall into regions of high overlap with the ID data (CIFAR-10). We define "overlap" and "separable" regions based on the distributions of the reconstruction error ($u(\mathbf{x})$) and the latent norm ($v(\mathbf{x})$), as visualized in Figure 4. This creates four scenarios of increasing difficulty:

- **Case 1 (Hardest):** Overlap in both reconstruction and latent spaces ($X_{\mathrm{overlap}} + Z_{\mathrm{overlap}}$).

- **Case 2:** Separable in latent space, overlap in reconstruction ($X_{\mathrm{overlap}} + Z_{\mathrm{separable}}$).

- **Case 3:** Overlap in latent space, separable in reconstruction ($X_{\mathrm{separable}} + Z_{\mathrm{overlap}}$).

- **Case 4 (Easiest):** Separable in both spaces ($X_{\mathrm{separable}} + Z_{\mathrm{separable}}$).

**Results.** The results, shown in Table 2, provide evidence consistent with likelihood cancellation and localize the regime in which it occurs. The partition is defined by the *marginal* separability of $u$ and $v$ and is distinct from the score being evaluated, so the comparison is not circular: "aggregate worse than best component" is not built into the partition. In the most challenging scenario, **Case 1**, the individual statistics $u(\mathbf{x})$ (AUROC 0.93) and $v(\mathbf{x})$ (AUROC 0.75) both contain a useful OOD signal, yet the ELBO that combines them degrades

to 0.83—below its own reconstruction term. Crucially, this degradation is *localized*: the ELBO is near-perfect where the reconstruction signal is separable (Cases 3–4: 1.00, 0.96) and collapses only in the overlap cases (Cases 1–2: 0.83, 0.87). A uniformly weak ELBO estimator, a sign error, or a normalization artifact would degrade it across all cases; and because AUROC is rank-based, monotone normalization cannot drive this pattern. The localized failure is thus the signature of destructive combination rather than these alternatives, though we do not claim to exclude every contributing factor (posterior-approximation error is itself part of the imperfect-model premise). Our LPath method, which combines the statistics with a classical detector, achieves AUROC 0.99—consistent with the LPath features retaining information from the full computational path that the final likelihood score loses. This addresses RQ1.

## 6    Conclusion

This work confronts a fundamental paradox in unsupervised OOD detection: the systematic failure of likelihood scores from powerful deep generative models. We identified a key mechanism for this failure in VAEs, a phenomenon we term **likelihood cancellation**, where informative OOD signals from the encoder and decoder neutralize each other in the final scalar likelihood.

To overcome this, we introduced the **Likelihood Path (LPath) Principle**, a new framework that moves beyond the final likelihood score to extract a richer signal from the model's entire computational path. We provided a multi-faceted analysis of this principle, grounding it in the statistical theories of likelihood and sufficiency, reinterpreting VAE inference through the neural lens of fast and slow weights, and demonstrating the cancellation effect from a combinatorial perspective.

Our resulting LPath method is principled, effective, and efficient. On standard benchmarks, our approach is competitive across the board and, on the hardest pairs, matches or exceeds methods that rely on models over ten times larger, at single-forward-pass cost. Using a lightweight 3M-parameter VAE, it offers a practical approach to real-world, streaming OOD detection where efficiency matters.

The LPath Principle offers a new perspective on OOD detection in generative models. Future work will involve extending this principled framework beyond VAEs to more powerful DGMs, such as normalizing flows and diffusion models, to explore its full potential as a general tool for building more reliable and robust machine learning systems.

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

## A    More Related Works

Prior works have approached OOD detection from various perspectives and with different data assumptions, e.g., with or without access to training labels, batches of test data, or single test data points in a streaming fashion, and with or without knowledge and inductive bias of the data. In the following, we give an overview organized by different data assumptions with a focus on where our method fits.

The first assumption is whether the method has access to training labels. There has been extensive work on classifier-based methods that assume access to training labels (Hendrycks & Gimpel, 2016; Frosst et al., 2019; Sastry & Oore, 2020; Bahri et al., 2021; Papernot & McDaniel, 2018; Osawa et al., 2019; Guénais et al., 2020; Lakshminarayanan et al., 2016; Pearce et al., 2020). Within this category, there are different assumptions as well, such as access to a pretrained network or knowledge of OOD test examples. See Table 1 of Sastry & Oore (2020) for a summary of such methods.

When we do not assume access to the training labels, the problem becomes more general and also harder. Under this category, some methods assume access to a batch of test data where either all the data points are OOD or not (Nalisnick et al., 2019). A more general setting does not assume OOD data would come in batches. Under this setup, there are methods that implicitly assume prior knowledge of the data, such as the input complexity method (Serrà et al., 2019), where the use of image compressors implicitly assumes an image-like structure, or the likelihood ratio method (Ren et al., 2019), where a noisy background model is trained with the assumption of a background-object structure.

**Relation to recent OOD work.**    Several adjacent lines of work operate under different assumptions. Supervised, classifier-based uncertainty methods—e.g. deterministic distance-aware models such as DUE (van Amersfoort et al., 2021) and SNGP (Liu et al., 2023a)—achieve single-forward-pass OOD detection but require class labels at training time, whereas our setting is strictly unsupervised, single-sample, and label-free; they address a different, label-rich problem. Recent work probing *intermediate representations* for OOD signals (Meza De la Jara et al., 2025) is closely related in spirit to the LPath view that internal computations are more informative than final outputs; that work studies discriminative, zero-shot vision encoders, whereas our contribution is the unsupervised *generative* instance, with a likelihood-path selection of statistics. Finally, broad benchmarks such as the generalized/full-spectrum and OpenOOD evaluations (Yang et al., 2021; 2023; Zhang et al., 2024) cover a wider range of distribution shifts than the unsupervised single-sample DGM-likelihood lineage we follow (Nalisnick et al., 2018; Xiao et al., 2020; Havtorn et al., 2021; Morningstar et al., 2021; Graham et al., 2023; Liu et al., 2023b); large-scale evaluation in that setting is clearly-scoped future work.

## B    Overcoming the Latent Dimensionality Trade-off

The 'LPath-2M' variant of our method is designed to resolve a fundamental trade-off inherent in using a single VAE for OOD detection. This section details this trade-off and explains how our two-model approach provides an effective solution.

**The Inherent Trade-off of a Single VAE.**    When using a single VAE, the choice of the latent dimension creates a conflict between the needs of the encoder and the decoder for effective OOD detection:

- **A Higher Latent Dimension Benefits the Encoder ($q_\phi$):** A large latent space provides the encoder with more capacity to capture complex features. This allows it to map in-distribution (ID) and OOD data to more distinct, separable regions, improving the discriminative power of statistics derived from the latent code (e.g., $v(\mathbf{x})$ and $w(\mathbf{x})$).

- **A Lower Latent Dimension Benefits the Decoder ($p_\theta$):** A small, bottlenecked latent space constrains the decoder's ability to generalize. It learns to reconstruct ID data well but struggles to accurately reconstruct unfamiliar OOD samples. This constraint is beneficial for OOD detection, as it leads to larger, more easily detectable reconstruction errors (i.e., a stronger signal from $u(\mathbf{x})$).

This trade-off means that any single choice of latent dimension is a compromise. Optimizing for the encoder's discriminative power (with a high dimension) weakens the decoder's reconstruction-based OOD signal, and vice-versa.

**The Two-Model Solution.** To overcome this trade-off, the 'LPath-2M' method uses two specialized VAEs in parallel:

1. **A High-Dimensional VAE:** This model has an overparameterized (large) latent dimension, optimized for the encoder. It excels at capturing complex features and producing informative latent statistics ($v(\mathbf{x})$ and $w(\mathbf{x})$) that effectively discriminate between ID and OOD data.

2. **A Low-Dimensional VAE:** This model has an underparameterized (small) latent dimension, optimized for the decoder. Its constrained nature amplifies reconstruction errors for OOD samples, providing a strong OOD signal via the decoder-based statistic $u(\mathbf{x})$.

By combining the complementary strengths of these two models, we can leverage the best of both worlds without being limited by the conflicting requirements of a single architecture.

**Implementation and Empirical Validation.** In practice, we implement the 'LPath-2M' method by extracting the LPath statistics from their respective specialist models:

- **From the High-Dimensional VAE**, we extract the encoder-based statistics:

$$v(\mathbf{x}) = \|\boldsymbol{\mu_z}(\mathbf{x})\|_2, \tag{17}$$
$$w(\mathbf{x}) = \|\boldsymbol{\sigma_z}(\mathbf{x})\|_2. \tag{18}$$

- **From the Low-Dimensional VAE**, we extract the decoder-based statistic:

$$u(\mathbf{x}) = \|\mathbf{x} - \widehat{\mathbf{x}}\|_2. \tag{19}$$

This integrated feature set provides a comprehensive signal for OOD detection. As shown in our main results (Table 1), this two-model approach is competitive on challenging OOD scenarios. This is achieved even though each VAE, considered individually, has potential limitations (e.g., the high-dimensional VAE may overfit, while the low-dimensional one may underfit). By combining their complementary strengths, the 'LPath-2M' method demonstrates that it can be more effective to use two specialized, simple models than a single, compromised one.

## C  Sufficient Statistics and Where to Find Them

For the Gaussian-parameterized VAE encoder and decoder, the parameters of the component distributions can be read off directly; the analogous quantities may be harder to identify for more complicated distributions. We emphasize, consistent with Section 4, that these component parameters are *not* classical minimal sufficient statistics for an unknown population parameter—we invoke sufficiency only as a classical analogy. Here we briefly overview the Fisher–Neyman factorization perspective on the classical sufficiency principle, which can help identify such quantities in more complicated distributions. A sufficiency statistics is also characterized by Fisher-Neyman factorization theorem (Wasserman, 2006): $T(\mathbf{x})$ is a sufficient statistics for $p(\mathbf{x}|\psi)$ parameterized by $\psi$ if and only if:

$$\ell(\psi|\mathbf{x}) = p(\mathbf{x}|\psi) = f(\mathbf{x})g_\psi(T(\mathbf{x})) \tag{20}$$

i.e. the density $p(\mathbf{x}|\psi)$ can be factored into a product such that $f$, does not depend on $\psi$ and $g$ that does depend on $\psi$ *but who depends on $\mathbf{x}$ only through $T(\mathbf{x})$*. For example, if we perform inference by maximum likelihood:

$$\psi_{\mathrm{MLE}} = \arg\max_{\psi} \ell(\psi|\mathbf{x}) = \arg\max_{\psi} f(\mathbf{x})g_{\psi}(T(\mathbf{x})) = \arg\max_{\psi} g_{\psi}(T) \tag{21}$$

$T$ is sufficient for MLE procedure, because $\psi_{\mathrm{MLE}}$ only requires $T$.

> The *sufficiency principle* states that, if $T(\mathbf{x})$ is a sufficient statistic for the likelihood function $p(\mathbf{x}|\psi)$, then any inference about $\psi$ should depend on $T(\mathbf{x})$ only.

## D  Experimental Details

### D.1  VAE Architecture and Training

For the architecture and the training of our VAEs, we followed Xiao et al. (2020). We have trained VAEs of varying latent dimensions, {1, 2, 5, 10, 100, 1000, 2000, 3096, 5000, 10000}; for each dimension, instead of training for a fixed 200 epochs and taking the final checkpoint, we kept the checkpoint with the best *in-distribution validation loss.*

**Model selection is in-distribution only.**  Both the training checkpoint *and* the latent dimension are selected by the same in-distribution validation-loss criterion; no out-of-distribution or test data is used at any point in model selection. For LPath-1M, the reported single VAE is the one minimizing in-distribution validation loss over all candidate latent dimensions. For LPath-2M, the high- and low-dimensional roles are fixed by the encoder/decoder trade-off of Appendix B—a property of the method, not a tuned hyperparameter— and within each role we again select the VAE with the best in-distribution validation loss: the high-dimensional model from {3096, 5000, 10000} (for the encoder statistics $v, w$) and the low-dimensional model from {1, 2, 5} (for the decoder statistics $u, s$).

In addition to Gaussian VAEs as mentioned in Section D.1, we also empirically experimented with a categorical decoder, in the sense the decoder output is between the discrete pixel ranges, as in Xiao et al. (2020). Strictly speaking, this is no longer a Gaussian distribution, which complicates the clean read-off of the component-distribution parameters our analysis relies on. We nonetheless experimented with it to test whether the LPath principle can serve as a heuristic to inspire methods in this non-Gaussian setting, and we observed that categorical decoders work similarly to Gaussian decoders.

**Constant Decoder Covariance**  In typical VAE learning, the decoder's variance is fixed (Dai & Wipf, 2019), so it cannot be used as an inferential parameter. We initially treated the decoder as an isotropic Gaussian with a learnable scalar covariance matrix $\sigma_{\mathbf{x}}(\mathbf{z})^2 I$, where $I$ is the identity matrix and $\sigma_{\mathbf{x}}(\mathbf{z})^2$ is a learnable scalar. We later observed that the scalar $\sigma_{\mathbf{x}}(\mathbf{z})$ always converge to a small value and remains fixed for any ID or OOD data. And given that in typical VAE learning, the decoder's variance is fixed (Dai & Wipf, 2019). We decided to use a fixed scalar as well and exclude this term from our algorithm.

This reduces the component parameters for the encoder and decoder pair:

$$(\mu_{\mathbf{z}}(\mathbf{x}), \sigma_{\mathbf{z}}(\mathbf{x}), \mu_{\mathbf{x}}(\mathbf{z}), \sigma_{\mathbf{x}}(\mathbf{z})) \longrightarrow (\mu_{\mathbf{z}}(\mathbf{x}), \sigma_{\mathbf{z}}(\mathbf{x}), \mu_{\mathbf{x}}(\mathbf{z})) \tag{22}$$

### D.2  Feature Processing To Boost COPOD Performances

Like most statistical algorithms, COPOD/MD is not scale invariant, and may prefer more dependency structures closer to the linear ones. When we plot the distributions of $u(\mathbf{x})$ and $v(\mathbf{x})$, we find that they exhibit extreme skewness. To make COPOD's statistical estimation easier, we process them by quantile transform. That is, for ID data, we map the the tuple of statistics' marginal distributions to $\mathcal{N}(0, 1)$. To ease the low dimensional empirical copula, we also de-correlate the joint distribution of $(u(\mathbf{x}), v(\mathbf{x})), w(\mathbf{x}))$. We do so using Kessy et al. (2018)'s de-correlation method, similar to Morningstar et al. (2021).

### D.3  Width and Height of a Vector Instead of Its $l^2$ Norm To Extract Complementary Information

In our visual inspection, we find that the distribution of the scalar components of $(u(\mathbf{x}), v(\mathbf{x}), w(\mathbf{x}))$ can be rather uneven. For example, the visible space reconstruction $\mathbf{x} - \widehat{\mathbf{x}}$ error can be mostly low for many

pixels, but very high at certain locations. These information can be washed away by the $l^2$ norm. Instead, we propose to track both $l^p$ norm and $l^q$ norm for small $p$ and large $q$.

**For small $p$, $l^p$ measures the width of a vector, while $l^q$ measures the height a vector for big $q$.** To get a sense of how they capture complementary information, we can borrow intuition from $l^p \approx l^0$, for small $p$ and $l^q \approx l^\infty$, for large $q$. $\|\mathbf{x}\|_0$ counts the number of nonzero entries, while $\|\mathbf{x}\|_\infty$ measures the height of $\mathbf{x}$. For $\mathbf{x}$ with continuous values, however, $l^0$ norm is not useful because it always returns the dimension of $\mathbf{x}$, while $l^\infty$ norm just measures the maximum component.

**Extreme measures help screen extreme data.** We therefore use $l^p$ norm and $l^q$ norm as a continuous relaxation to capture this idea: $l^p$ norm will "count" the number of components in $\mathbf{x}$ that are unusually small, and $l^q$ norm "measures" the average height of the few biggest components. These can be more discriminative against OOD than $l^2$ norm alone, due to the extreme (proxy for OOD) conditions they measure. We observe some minor improvements, detailed in Table 4's ablation study.

| ID: CIFAR10 | | OOD | | |
|---|---|---|---|---|
| OOD Dataset | SVHN | CIFAR100 | Hflip | Vflip |
| $l^2$ norm | 0.96 | 0.60 | **0.53** | **0.61** |
| $(l^p, l^q)$ | **0.99** | **0.62** | **0.53** | **0.61** |

Table 4: Comparing the AUC of $l^2$ norm versus our $(l^p, l^q)$ measures.

### D.4 (Attempted) Training Objective Modification for Stronger Concentration

Inspired by the well known concentration of Gaussian probability measures, to encourage stronger concentration of the latent code around the spherical shell with radius $\sqrt{m}$ for better OOD detection, we propose the following modifications to standard VAEs' loss functions:

We replace the initial KL divergence by:

$$\mathcal{D}^{\text{typical}}[Q_\phi(\mathbf{z} \mid \mu_{\mathbf{z}}(\mathbf{x}), \sigma(\mathbf{x}))\|P(\mathbf{z})] \tag{23}$$

$$=\mathcal{D}^{\text{typical}}[\mathcal{N}(\mu_{\mathbf{z}}(\mathbf{x}), \sigma_{\mathbf{z}}(\mathbf{x}))\|\mathcal{N}(0, I)] \tag{24}$$

$$=\frac{1}{2}\left(\text{tr}(\sigma_{\mathbf{z}}(\mathbf{x})) + |(\mu_{\mathbf{z}}(\mathbf{x}))^\top (\mu_{\mathbf{z}}(\mathbf{x})) - m| - m - \log\det(\sigma_{\mathbf{z}}(\mathbf{x}))\right) \tag{25}$$

where $m$ is the latent dimension.

In training, we also use Maximum Mean Discrepancy (MMD) Gretton et al. (2012) as a discriminator since we are not dealing with complex distribution but Gaussian. The MMD is computed with Gaussian kernel. This extra modification is because the above magnitude regularization does not take distribution in to account.

The final objective:

$$\mathbb{E}_{\mathbf{x}\sim P_{\text{ID}}}\mathbb{E}_{\mathbf{z}\sim Q_\phi}\mathbb{E}_{\mathbf{n}\sim\mathcal{N}}[\log P_\theta(\mathbf{x} \mid \mathbf{z}) - \mathcal{D}^{\text{typical}}[Q_\phi(\mathbf{z} \mid \mu_{\mathbf{z}}(\mathbf{x}), \sigma(\mathbf{x}))\|P(\mathbf{z})] - \text{MMD}(\mathbf{n}, \mu_{\mathbf{z}}(\mathbf{x})) \tag{26}$$

The idea is that for $P_{\text{ID}}$, we encourage the latent codes to concentrate around the prior's *typical sets*. That way, $P_{\text{OOD}}$ may deviate further from $P_{\text{ID}}$ in a controllable manner. In experiments, we tried the combinations of the metric regularizer, $\mathcal{D}^{\text{typical}}$, and the distribution regularizer, MMD. This leads to two other objectives:

$$\mathbb{E}_{\mathbf{x}\sim P_{\text{ID}}}\mathbb{E}_{\mathbf{z}\sim Q_\phi}[\log P_\theta(\mathbf{x} \mid \mathbf{z}) - \mathcal{D}^{\text{typical}}[Q_\phi(\mathbf{z} \mid \mu_{\mathbf{z}}(\mathbf{x}), \sigma(\mathbf{x}))\|P(\mathbf{z})] \tag{27}$$

$$\mathbb{E}_{\mathbf{x}\sim P_{\text{ID}}}\mathbb{E}_{\mathbf{z}\sim Q_\phi}\mathbb{E}_{\mathbf{n}\sim\mathcal{N}}[\log P_\theta(\mathbf{x} \mid \mathbf{z}) - \mathcal{D}[Q_\phi(\mathbf{z} \mid \mu_{\mathbf{z}}(\mathbf{x}), \sigma(\mathbf{x}))\|P(\mathbf{z})] - \text{MMD}(\mathbf{n}, \mu_{\mathbf{z}}(\mathbf{x})) \tag{28}$$

where $\mathcal{D}$ is the standard KL divergence.

But we **did not** observe a significant difference in the final AUROC different variations. We still include those attempted modifications for future work.

| Statistic | OOD Dataset | | | |
|---|---|---|---|---|
| | SVHN | CIFAR100 | Hflip | Vflip |
| $u(\mathbf{x})$ | 0.96 | 0.59 | 0.54 | 0.59 |
| $v(\mathbf{x})$ | 0.94 | 0.56 | 0.54 | 0.59 |
| $w(\mathbf{x})$ | 0.93 | 0.58 | 0.54 | 0.61 |
| $v(\mathbf{x})$ & $w(\mathbf{x})$ | 0.94 | 0.58 | 0.54 | 0.60 |
| $u(\mathbf{x})$ & $v(\mathbf{x})$ | 0.97 | 0.61 | 0.53 | 0.61 |
| $u(\mathbf{x})$ & $w(\mathbf{x})$ | 0.98 | 0.61 | 0.54 | 0.61 |

Table 5: COPOD on individual statistics. ID dataset is CIFAR10.

## E  Ablation Studies

### E.1  Individual statistics

To empirically validate how $(u(\mathbf{x}), v(\mathbf{x}), w(\mathbf{x}))$ complement each other, we use individual component alone in first stage and fit the second stage COPOD as usual. We notice significant drops in performances. We fit COPOD on individual statistics $u(\mathbf{x})$, $v(\mathbf{x})$, $w(\mathbf{x})$ and show the results in Table 5. We can see that our original combination in Table 1 is better overall.

### E.2  MD

To test the efficacy of $(u(\mathbf{x}), v(\mathbf{x}), w(\mathbf{x}))$ without COPOD, we replace COPOD by a popular algorithm in OOD detection, the MD algorithm Lee et al. (2018) and report such scores in Table 1. The scores are comparable to COPOD, suggesting $(u(\mathbf{x}), v(\mathbf{x}), w(\mathbf{x}))$ is the primary contributor to our performances.

### E.3  Latent dimensions

One hypothesis on the relationship between latent code dimension and OOD detection performance is that lowering dimension incentivizes high level semantics learning, and higher level feature learning can help discriminate OOD v.s. ID. We conducted experiments on the below latent dimensions and report their AUC based on $v(\mathbf{x})$ (norm of the latent code) in Table 6

| Latent dimension | 1 | 2 | 5 | 10 | 100 | 1000 | 3096 | 5000 |
|---|---|---|---|---|---|---|---|---|
| $v(\mathbf{x})$ AUC | 0.39 | 0.63 | 0.52 | 0.45 | 0.22 | 0.65 | 0.76 | 0.59 |

Table 6: Lower latent code dimension doesn't help to discriminate in practice.

Clearly, lowering the dimension isn't sufficient to increase OOD performances.

## F  Reproducibility

This appendix consolidates the information needed to reproduce our method and experiments.

**Model.**  We use a Gaussian VAE with the DC-VAE architecture and training setup of Xiao et al. (2020) (approximately 3M parameters). The prior is $p(\mathbf{z}) = \mathcal{N}(\mathbf{0}, \mathbf{I})$, the encoder is $q_\phi(\mathbf{z} \mid \mathbf{x}) = \mathcal{N}(\boldsymbol{\mu}_z(\mathbf{x}), \text{diag}(\boldsymbol{\sigma}_z^2(\mathbf{x})))$, and the decoder is a Gaussian $p_\theta(\mathbf{x} \mid \mathbf{z}) = \mathcal{N}(\boldsymbol{\mu}_x(\mathbf{z}), \sigma_x^2 \mathbf{I})$ whose covariance is a fixed scalar (Appendix D.1). We follow Xiao et al. (2020) for the training hyperparameters; for each model we retain the checkpoint with the best in-distribution validation loss rather than the final-epoch checkpoint.

**Latent dimensions and model selection.**  We trained VAEs at latent dimensions $\{1, 2, 5, 10, 100, 1000, 2000, 3096, 5000, 10000\}$. *All model selection uses in-distribution validation loss only; no out-of-distribution or test data is used at any stage* (Appendix D). LPath-1M uses the single

VAE that minimizes in-distribution validation loss over all candidate dimensions. LPath-2M assigns a high-dimensional VAE to the encoder statistics $(v, w)$ and a low-dimensional VAE to the decoder statistic $u$, following the trade-off of Appendix B; within each role we again take the VAE with the best in-distribution validation loss, drawn from the high-dimensional group $\{3096, 5000, 10000\}$ and the low-dimensional group $\{1, 2, 5\}$ respectively.

**Stage 1: feature extraction.** From a trained VAE we compute the LPath statistics $u(\mathbf{x}) = \|\mathbf{x} - \boldsymbol{\mu}_x(\boldsymbol{\mu}_z(\mathbf{x}))\|_2$, $v(\mathbf{x}) = \|\boldsymbol{\mu}_z(\mathbf{x})\|_2$, and $w(\mathbf{x}) = \|\boldsymbol{\sigma}_z(\mathbf{x})\|_2$ (Section 3.3.2). Because the decoder covariance is a fixed scalar, the reconstruction-standard-deviation statistic $s(\mathbf{x})$ is constant and is dropped, so the implemented feature vector is the three-dimensional $(u, v, w)$. Optionally, the $\ell^2$ norm is replaced by small-$p$ and large-$q$ norms to retain coordinate-wise information (Appendix D.3), which gives a small improvement (Table 4).

**Stage 2: classical detector.** The features are scored by a classical density estimator fit on the in-distribution features: COPOD (Li et al., 2020) or a Mahalanobis-distance (MD) detector (Lee et al., 2018). For COPOD we apply the standard preprocessing of a quantile transform—mapping each in-distribution marginal to $\mathcal{N}(0, 1)$—followed by decorrelation (Kessy et al., 2018), as in Morningstar et al. (2021) (Appendix D.2).

**Datasets and metric.** In-distribution datasets are CIFAR-10, SVHN, FMNIST, and MNIST; the OOD sets are the remaining datasets together with CIFAR-100 and horizontally/vertically flipped (Hflip/Vflip) variants. Detection performance is measured by AUROC.

