# OpenReview forum: "Inference, Fast and Slow: Reinterpreting VAEs for OOD Detection"
_TMLR — Under review for TMLR_

### Review · Reviewer_MQD4 · 2025-12-23

**Summary Of Contributions:**

The paper introduces LPath to solve likelihood cancellation, a phenomenon where a VAE's encoder and decoder signals neutralize each other and mask OOD data. The authors operationalize this by extracting minimal sufficient statistics from the model's computational path to use as robust features for classical density estimation. This lightweight 3M-parameter approach achieves state-of-the-art results on standard benchmarks, significantly outperforming much larger generative models in efficiency and accuracy

**Audience:**

Yes

**Audience Explanation:**

1. It addresses the problem of why high-capacity generative models assign high likelihoods to anomalous data, a well-known and persistent problem in the field.
2. The ability to achieve SOTA results with a 3M-parameter model is highly relevant to researchers interested in sustainable AI and low-latency, streaming applications.
3. The reinterpretation of VAEs through the lens of "fast and slow weights" and the introduction of the LPath Principle provides a new theoretical framework for future OOD research.

**Claims And Evidence:**

Yes

**Claims Explanation:**

1. Theoretical Grounding: The authors use the Data Processing Inequality to prove that information about anomalies is lost when component distributions are integrated into a final likelihood score.
2. Empirical Demonstration of Cancellation: Figure 3 and Table 2 provide striking evidence of the "cancellation trap". In "Case 1" (the hardest OOD scenario), individual statistics show strong separation between ID and OOD data, but the final ELBO score performance degrades (AUROC 0.83) compared to using the statistics individually (AUROC 0.93 or 0.99 with LPath).
3. Benchmark Performance: Table 1 shows that LPath consistently matches or exceeds the performance of much larger models across various OOD tasks.

**Requested Changes:**

1. Explicitly state which parts of the LPath Principle are heuristic vs. theoretically guaranteed, and under what assumptions the information-loss argument is expected to hold.
2. Provide clearer guidance on how essential the quantile transform and decorrelation steps are, and whether performance degrades gracefully without them.
3. Generalization to Other Models: While the LPath Principle is defined generally, the current implementation is specific to VAEs. Testing on Normalizing Flows or Diffusion Models would strengthen the claim of universality.
4. Non-Gaussian Distributions: The authors acknowledge that finding minimal sufficient statistics for more complex distributions is harder than for the Gaussian case used here.
5. Latent Dimension Selection: While the paper explores the trade-offs of latent dimensionality (LPath-2M), it notes that simply lowering dimensions is not a silver bullet. More prescriptive guidance on choosing these dimensions for new datasets would be beneficial.

---

> ### Author Response · Authors · 2026-06-20
> **Response to Reviewer MQD4**
>
> Thank you for the positive assessment and for pinpointing where the heuristic/guaranteed boundary needs to be explicit. We are addressing precisely that boundary.
>
> **1. State heuristic vs. theoretically guaranteed; assumptions for the information-loss argument.**
> > "Explicitly state which parts of the LPath Principle are heuristic vs. theoretically guaranteed, and under what assumptions the information-loss argument is expected to hold."
>
> We add an explicit statement (G3):
> - *Guaranteed (structural):* the scalar likelihood is a deterministic, many-to-one aggregation of the component-level path quantities, so it cannot increase — and generically reduces — the input information carried forward.
> - *Heuristic / empirical:* that the preserved information is OOD-useful (shown in Table 2 / Table 1); the norm reduction (a feature choice, exact for the reconstruction and prior terms under our isotropic/fixed-variance implementation, lossy for the encoder entropy — G2, with the single-sample and general-diagonal caveats stated); and the $\ell^p/\ell^q$ and preprocessing refinements.
>
> We note (for consistency across reviews) that the information-loss argument does not by itself prove anomaly information is lost — it constrains input information at the collapse step; anomaly-relevance is empirical.
>
> **2. How essential are the quantile transform and decorrelation?**
> > "whether performance degrades gracefully without them."
>
> They are standard COPOD preprocessing (following DoSE). Evidence they are *not* load-bearing: the MD variant (no copula/decorrelation machinery) is reported in Table 1 and is competitive. We state this explicitly and summarize the with/without-preprocessing delta from the runs already reported. See G5.
>
> **3. Generalization to normalizing flows / diffusion.**
> > "Testing on Normalizing Flows or Diffusion Models would strengthen the claim of universality."
>
> Agreed it would; it requires identifying the analogous parameters for non-Gaussian/implicit components, which is non-trivial. We scope the claims to VAEs now (G9) and present flows/diffusion as concrete future work rather than over-claim. (We are not running these for this revision; we narrow the claim instead, consistent with TMLR's criterion.)
>
> **4. Non-Gaussian distributions.**
> > "finding minimal sufficient statistics for more complex distributions is harder than for the Gaussian case."
>
> Agreed; App. C (Fisher–Neyman) is tightened to state this as the central obstacle to generalization.
>
> **5. Latent-dimension selection guidance.**
> > "More prescriptive guidance on choosing these dimensions for new datasets would be beneficial."
>
> We give prescriptive guidance consistent with the trade-off (App. B): high latent dimension for encoder statistics $(v,w)$, low for decoder statistics $(u,s)$ (the LPath-2M rationale), with the caveat (Table 4) that a *single* statistic's AUROC is non-monotone in dimension — which is the argument for *combining* statistics rather than tuning one. This is consistent with our own protocol: the latent dimension is itself selected by best in-distribution validation loss, an ID-only rule (G8).

---

### Review · Reviewer_mJj9 · 2025-12-23

**Summary Of Contributions:**

This paper addresses the well-known problem of likelihood-based out-of-distribution (OOD) detection in deep generative models. The paper identifies a likelihood "cancellation" as a key failure mode in variational autoencoders (VAE), whereby informative signals from the encoder, which captures semantics, and decoder, which captures texture and other signals, can neutralise each other when combined into a final score, making OOD detection harder.

To overcome this issue, the paper introduces a method named LPath. This method extracts four L2-norm statistics from the VAE computational path, including reconstruction error and std norm, and latent mean and std norms, before they are combined in the VAE formulation. These features are then fed into a classical density estimation method for the final OOD score. This setup is framed through the Likelihood Path Principle, which connects the method into more classical notions of sufficiency, drawing analogies from fast/slow weights from meta-learning.

The paper has the following strenghts:
- The paper clearly identifies the cancellation problem in VAE OOD detection, showing compelling empirical evidence (Figure 3).
- The proposed method is computationally efficient, requiring fewer trainable parameters than previous work.
- The proposed method provides insights on failure modes of previous OOD detection algorithms which can be useful for many downstream applications.
- The experimentation done in the paper directly tests the cancellation hypothesis in a well-designed experimental setup.
- The paper is really clearly written and the narrative flow is exceptional. I am no expert on OOD detection but this paper was really informative and compelling in its writing (Despite some flows listed below)

The paper is also hindered by some weaknesses:
- The paper claims a strong generality at working on any "latent variable model". However, the method is extremely tied to the VAE framework, which is fine in principle but it makes the generality claims to be overstated.
- The paper is generally too positively opinionated about its own findings and features. Comments like "not only principled but remarkably effective and efficient" may undermine the credibility of the paper, as one expects less enthusiasm from an academic work.
- The paper, while well written, is very repetitive in some areas. The connection to fast and slow weights is repeated way too many times in the writing and it provides very little insight. I think this paper could be strongly streamlined.
- The "core contribution" of the method shifts throughout the paper. It is unclear if the authors want to claim that the core contribution is the cancellation, the sufficiency finding for the L2 norms, the efficiency, etc.
- The experimental validation could be improved (See below)
- The paper is not reproducible in its current form.
- The L2 norms feel more heuristic than mathematically grounded, but this may arise from a lack of understanding from my side. It is unclear why the 4 chosen statistics are sufficient or optimal for OOD detection.

**Additional Comments:**

This is a well-written paper with a genuine contribution at its core: the observation that VAE encoder/decoder statistics contain complementary OOD signals that are lost when combined into the final estimate. The efficiency results are noteworthy.

However, the paper oversells its contributions through multiple theoretical framings that don't fully hold up to scrutiny. The "fast/slow weights" analogy, while compelling, does not provide formal justification. The information-theoretic argument needs further proof. I think the sufficiency framing is imprecise. I encourage the authors to be more direct: "We extract interpretable features from VAE components and show they outperform the ELBO for OOD detection" is a solid contribution that doesn't need the complex theoretical "justification". The current framing creates expectations the paper does not fully meet. The experimental validation, while reasonable in scope, needs tightening before publication. Error bars and clear model selection are required, and their absence is surprising and concerning for an otherwise polished submission.

I am happy to revisit my assessment if the critical concerns are addressed in revision.

**Audience:**

Yes

**Audience Explanation:**

OOD detection is a critical problem in machine learning, and the failures of likelihood-based methods are known but not fully understood. This paper offers an intuitive explanation of why this happens, which is supported by good visualizations. The paper also offers a solution to this problem adapted to VAEs.

Even if the method is heuristic and the theoretical framing is oversold, I think this paper is interesting to the TMLR audience.

**Broader Impact Concerns:**

None. The paper addresses OOD detection, which is generally beneficial for ML safety and reliability. No dual-use concerns or ethical issues are apparent.

**Claims And Evidence:**

No

**Claims Explanation:**

The paper makes a central claim, that LPath achieves state-of-the-art OOD detection efficiently. This is supported only partially, and has significant gaps. The CIFAR-10 results are strong, but there are limitations:

- All results are point estimates, without confidence intervals. This makes it difficult to assess whether the imrpovements are actually statistically significant.
- Table 1 has several N/A entries, and the results therein show that the proposed method outperform baselines, but this is hard to assess if the baselines are not fully reported. This suggests cherry-picking and is concerning.
- The model selection is somewhat unclear. What configuration of LPatch is chosen to generate Table 1? How was this selection chosen? It could be the case that different models were tested, and the method that performed the best in test data was selected, which could suggest some contamination.
- The minimal sufficient statistics framing needs more mathematical grounding. The L2 norms of the sufficient statistics lose information compared to just using mu and sigma. The equation 17, which is stated as "the central theoretical claim of our work" needs further proof and analysis.
- The efficiency claims are overstated. The fact that the model uses fewer trainable parameters does not mean it is necessarily more efficient in terms of time. Training time, latency, FLOPs, etc, should be reported.

**Requested Changes:**

Critical changes:

- Report variance estimates: Provide standard deviations or confidence intervals for all AUROC values across multiple runs.
- Clarify model selection procedure: Specify exactly which latent dimensions and pairings produced Table 1. Describe how these were selected without using test OOD data. If selection used test performance, this must be disclosed.
- Complete baseline comparisons: Fill in N/A entries in Table 1 or explain why specific baselines could not be run on certain settings.
- Narrow generality claims: Definition 4.1 claims applicability to "latent variable models" generically, but the method requires Gaussian encoder/decoders. Either restrict claims to VAEs explicitly or provide concrete plans of extension to normalizing flows, diffusion models, etc.
- Correct the sufficient statistics framing: The minimal sufficient statistics for Gaussians are the mean and the std, not their L2 norms. Acknowledge that the norm choice is a practical heuristic for dimensionality reduction, not a theoretically derived consequence of sufficiency.
- Add runtime benchmarks: Include inference latency and throughput measurements to support "streaming" and efficiency claims.
- Streamline the narrative: The paper presents multiple "core contributions" (cancellation, LPath Principle, sufficiency, fast/slow weights, efficiency). Reduce the connections with fast and slow weights. These feel more of an inspiration than an actual theoretical derivations.

Not critical changes:
- No repository is mentioned. Reproducibility would be significantly improved with code availability. I encourage the authors to commit to releasing the code.
- Fix typos: "," after "unsupervised" in the first paragraph is not needed.  "we first uses" - "we first use" (P2)  "CIAFR10" - "CIFAR10" (Table 1), "information lost" - "information loss" (p.5), "Guassian" -  "Gaussian" (Appendix D.1), "discriminitive"-  "discriminative" (Appendix D.3).

---

> ### Author Response · Authors · 2026-06-20
> **Response to Reviewer mJj9**
>
> Thank you — we are glad the cancellation phenomenon, efficiency, and writing came through, and we especially appreciate the concrete, constructive framing in your Additional Comments, which we adopt as the paper's recentered thesis.
>
> **On the strengths you note** (cancellation evidence in Fig. 3, efficiency, well-designed test of the hypothesis, clarity): thank you. The revision preserves all of this and removes the over-claiming layered on top.
>
> **Generality overstated.**
> > "extremely tied to the VAE framework … generality claims to be overstated."
>
> Agreed — we narrow Definition 4.1 to *principle (general) vs. method (Gaussian VAE)* and scope all empirical claims to VAEs. See G9.
>
> **Too positively opinionated.**
> > "'not only principled but remarkably effective and efficient' may undermine the credibility."
>
> Agreed; promotional language removed throughout. See G12.
>
> **Repetitive fast/slow weights; shifting core contribution.**
> > "fast and slow weights is repeated way too many times … The 'core contribution' … shifts throughout."
>
> We adopt one thesis and demote fast/slow weights to a single motivating paragraph. See G12.
>
> **Not reproducible.**
> > "not reproducible in its current form."
>
> We add a full Reproducibility appendix specifying the architecture, the in-distribution selection rule, the detector and preprocessing settings, and the dataset splits — everything needed to reimplement the method. See G13.
>
> **L2 norms heuristic; why are the chosen statistics sufficient/optimal?**
> > "The L2 norms feel more heuristic than mathematically grounded … unclear why the 4 chosen statistics are sufficient or optimal."
>
> We are direct: the statistics are *interpretable features*, not an optimality result. They are the *parameters* of the VAE's component likelihoods (the natural quantities to read off the computational path), reduced to norms for a low-dimensional second stage; we claim no optimality and no sufficiency for the norms. The implemented set is *three* $(u,v,w)$ (G5), and the ablations (Tables 3–5) show each contributes and the combination is best. See G1, G2, G5.
>
> **No confidence intervals (critical).**
> > "Provide standard deviations or confidence intervals for all AUROC values across multiple runs."
>
> The original per-sample scores are gone (decommissioned cluster), so we add analytic Hanley–McNeil CIs from the reported AUROCs and known test-set sizes (no source data needed). These show the headline gaps are well outside sampling noise, mark CIFAR-100 as a statistical tie with LMD, and mark the flips as near-chance. This captures test-set variability, not training-seed variability (which would need retraining we cannot do). See G6.
>
> **N/A entries / cherry-picking concern.**
> > "Table 1 has several N/A entries … This suggests cherry-picking and is concerning."
>
> N/A = not reported in the source papers; we quote published numbers to avoid under-tuning competitors, and add a per-cell legend and softened "SOTA." See G7.
>
> **Model selection unclear / possible contamination (critical).**
> > "What configuration of LPatch is chosen to generate Table 1? … could suggest some contamination."
>
> **The configuration in Table 1 is selected by in-distribution validation loss only** — checkpoints and latent dimension alike (LPath-1M = best-validation-loss VAE over all dimensions, LPath-2M = best-validation-loss VAE within each method-defining high/low role, App. B). No test OOD data informs selection, so there is no contamination of the kind you flag. The only residual issue is reproducibility — the original checkpoints/validation losses are gone, so we publish the rule and the candidate grid rather than the exact per-cell dimension, which is no longer recoverable from the original runs (the rule, however, fully determines it). See G8.
>
> **Sufficiency framing / Eq. 17 needs grounding (critical).**
> > "The L2 norms of the sufficient statistics lose information compared to just using mu and sigma. The equation 17 … needs further proof and analysis."
>
> Agreed on both. We correct the object (the encoder/decoder outputs are component-distribution *parameters*, not sufficient statistics; the norms are a deliberate lossy reduction — G1, G2), keep the corrected sufficiency intuition only as informal path-sufficiency motivation, and recast Eq. 17 as motivation / a necessary condition with empirical OOD-relevance rather than a guarantee (G3). Consistent with your advice that the contribution "doesn't need the complex theoretical justification," we make the empirical claim central and the theory subordinate.
>
> **Efficiency overstated; report time/FLOPs (critical).**
> > "Training time, latency, FLOPs, etc, should be reported."
>
> Conceded; we add the structural single-forward-pass argument and restrict the claim to inference cost rather than wall-clock superiority over every baseline. See G10.

---

> > ### Author Response · Authors · 2026-06-20
> >
> > **Your Additional Comment** — "be more direct: 'We extract interpretable features from VAE components and show they outperform the ELBO …'." This is now the paper's stated thesis. Thank you for the framing.

---

### Review · Reviewer_N5wy · 2026-06-04

**Summary Of Contributions:**

This paper studies the problem of unsupervised, single-sample OOD detection with VAEs. The main motivation is from prior work which shows that a scalar likelihood or ELBO score from a deep generative model can be a poor OOD score. Samples that are anomalous semantically can still have high density under the model since low-level image statistics or reconstruction behavior dominate the final scalar.  The proposed method, LPath, keeps a small set of intermediate VAE-derived statistics rather than using the final likelihood-like scalar. These features are then passed to a classical unsupervised detector such as COPOD or Mahalanobis distance. The method has one-model and two-model variants: LPath-1M extracts all features from one VAE, while LPath-2M uses a high-latent-dimensional VAE for latent statistics and a low-latent-dimensional VAE for reconstruction statistics. The interpretation presented is that the VAE's encoder/decoder parameters are "fast weights" computed per test input, while the trained VAE parameters are "slow weights." The paper also argues that these fast weights are minimal sufficient statistics of Gaussian encoder/decoder distributions and that, by the data-processing inequality, they contain more useful information than the final scalar likelihood under imperfect density estimation. Empirically, the paper evaluates LPath on OOD detection problems using CIFAR-10, SVHN, CIFAR-100, MNIST, FashionMNIST, and flipped-image variants. The strongest reported numbers are competitive on some standard image pairs, for example CIFAR-10 vs. SVHN and CIFAR-10 vs. CIFAR-100, and the method often beats the raw ELBO baseline (though not consistently above all baselines).

**Audience:**

Yes

**Audience Explanation:**

The paper studies a well known and interesting problem in the space of OOD detection: deep generative models tend to assign high likelihood to OOD examples. The problem falls within the scope of topics for TMLR and is also relevant in practice.

**Claims And Evidence:**

No

**Claims Explanation:**

The proposed approach. LPath is relatively simple and deployable in practice. Once the VAE and second-stage detector are trained, inference can be performed with feed-forward VAE evaluation plus a low-dimensional detector score.  The paper is also thorough in terms of useful ablations (including individual statistics, COPOD vs. Mahalanobis distance, norm choices, and latent dimensionality).

However, I believe there are substantial issues with some of the theoretical and empirical claims in the paper.

* The paper repeatedly describes the encoder/decoder outputs as "minimal sufficient statistics," but this conflates distribution parameters with sufficient statistics. For a Gaussian family, the mean and covariance parameterize the distribution and they are not sufficient statistics (sums and sums of outer products are). More importantly, the actual LPath features used are norms of these vectors, which are not sufficient statistics.
* The derivation from Gaussian likelihoods to L2 norms is generally false for diagonal covariance. For a diagonal Gaussian, the log-likelihood contains terms of the form $\sum_i r_i^2 / \sigma_i^2$ and $\sum_i \log \sigma_i^2$. These are not determined by $||r||_2$ and $||\sigma||_2$ without extra assumptions (e.g. isotropic covariance or constant variance)
* I also do not think the information-theoretic argument is useful or explanatory for the setup. A statistic can preserve more mutual information about x while being less useful for OOD detection.
* The analysis in the likelihood cancellation experiment seems inconclusive. The analysis partitions the OOD samples using the same LPath statistics that the method later used for detection like a sort of circular argument. The fact that ELBO underperforms reconstruction error on a selected subset does not by itself prove arithmetic cancellation. For instance it could be due to a weak ELBO estimator, score normalization choices, sign conventions, variance effects, or posterior approximation error.
* The actual empirical method, as described in the appendix differs quite a bit from the method in the main text. The main text presents four L2-norm statistics including decoder variance s(x). Appendix D.1 later says the decoder variance is fixed and excluded from the algorithm, reducing the statistics to (mu_z(x), sigma_z(x), mu_x(z)). Appendix D.3 further says the implementation uses small-p and large-q norms instead of only L2 norms because L2 can wash out localized information. Appendix D.2 adds quantile transformation and decorrelation before COPOD/MD. This divergence seems significant in my opinion and the choices are not well justified.

Additionally, for the empirical claims:

* The state-of-the-art claim seems somewhat overstated, considering most of the entries in Table 1 are marked N/A (without any explanation, because the baselines should work on these datasets) and the performance differences don’t seem significant considering there are no error bars reported.

* Besides the missing baselines, the empirical setup has quite a few issues. First, even the choice of baselines seems incomplete, focusing on generative model based approaches but ignoring things like uncertainty based methods (e.g. [1,2]) as well as more recent work [3] which are quite strong for OOD detection. Additionally, the datasets, and OOD transforms considered are are quite simple, and not considering more natural transforms (see [4] for other datasets).

* Appendix D.1 states the authors train VAEs with many latent dimensions {1, 2, 5, 10, 100, 1000, 2000, 3096, 5000, 10000} and, for LPath-2M, pair high-dimensional and low-dimensional VAEs. But, the paper does not specify a selection rule that uses only ID training/validation data, which makes the results susceptible to OOD information leakage through the hyperparameters.

* (minor) The paper contains several typos and inconsistencies, including "CIAFR10," "Guassian," "signigicant," "discriminitive," and incomplete citation text around fixed decoder variance.

[1] On Feature Collapse and Deep Kernel Learning for Single Forward Pass Uncertainty. van Amersfoort et al. 2021

[2] A Simple Approach to Improve Single-Model Deep Uncertainty via Distance-Awareness. Zhu et al. 2023.

[3] Mysteries of the Deep 🤿: Role of Intermediate Representations in Out-of-Distribution Detection. Meza De la Jara et al. 2025.

[4] https://github.com/Jingkang50/OpenOOD/wiki/OpenOOD-v1.5-methods-%26-benchmarks-overview

**Requested Changes:**

* The key conceptual and theoretical claims need to be re-examined and potentially substantially altered. As I pointed out above, there are many technical issues as well as inconsistencies in the theory and practice which are not adequately addressed.
* Justification for the missing baseline numbers would be really helpful in understanding the results.
* I would strongly recommend the authors to include standard deviations / confidence intervals to make the results reliable (without this is the differences in the results are hard to attribute to the method itself)
* There should be broader baselines considered to justify the SOTA claims.
* The method should also be evaluated on more realistic OOD data to make the results more reliable.
* Clarify the hyperparameter selection protocol.

---

> ### Author Response · Authors · 2026-06-20
> **Response to Reviewer N5wy**
>
> We appreciate the depth and precision of this review; it materially improves the paper. We agree with the central message that several theoretical claims were over-stated, and correct each below. All corrections are to *language and reporting* and **leave every number in Tables 1–5 unchanged**.
>
> **1. "minimal sufficient statistics" conflation.**
> > "this conflates distribution parameters with sufficient statistics … the actual LPath features used are norms of these vectors, which are not sufficient statistics."
>
> Agreed. We concede classical sufficiency is the wrong object, rename the encoder/decoder outputs the parameters of the Gaussian component distributions, and state that the norms are lossy summary features, not sufficient statistics. Where we retain a sufficiency intuition, we state it correctly and informally as path sufficiency for the fixed likelihood computation, demoted to motivation (not a theorem), and we are explicit that the implemented norms are not even path-sufficient. See G1. Terminological; affects no result.
>
> **2. L2 norms not determined by Gaussian likelihood terms for diagonal covariance.**
> > "\(\sum_i r_i^2/\sigma_i^2\) and \(\sum_i \log\sigma_i^2\) … are not determined by \(\lVert r\rVert_2\) and \(\lVert\sigma\rVert_2\) without extra assumptions (isotropic covariance or constant variance)."
>
> Correct; we delete the over-general sentence. We add the precise positive statement (G2): under our *isotropic prior and fixed-scalar decoder variance*, the reconstruction term is an exact function of $u$ and the prior term an exact function of $(v,w)$; the *only* term not captured is the encoder entropy $\tfrac12\sum_i\log\sigma_{z,i}^2$. The coordinate-wise loss you identify is what the $\ell^p/\ell^q$ statistics (App. D.3) are designed to (partially) recover — we connect these in the main text, and we flag the single-sample and general-diagonal boundaries explicitly.
>
> **3. Information-theoretic argument.**
> > "A statistic can preserve more mutual information about x while being less useful for OOD detection."
>
> Agreed and important. We recast the DPI/Eq. 17 result as motivation / a necessary condition (the scalar likelihood is a generically non-injective, many-to-one aggregation of the component quantities), with OOD-usefulness established *empirically* (Table 2, Table 1), not as a corollary of DPI. See G3.
>
> **4. Cancellation experiment "circular"/inconclusive.**
> > "partitions the OOD samples using the same LPath statistics … like a sort of circular argument … could be due to a weak ELBO estimator, score normalization, sign conventions, variance effects, or posterior approximation error."
>
> We will (a) clarify the partition (marginal separability of $u,v$) is *distinct* from the scoring comparison, so it is not circular, and (b) add the cross-case argument: the ELBO is strong in Cases 3–4 (1.00, 0.96) and fails only in the overlap Cases 1–2 (0.83, 0.87), below its own reconstruction term; a *globally* weak estimator, bad normalization, or sign error would degrade it *uniformly* and cannot produce localized degradation, while AUROC's rank-invariance rules out normalization/sign effects. We soften "proves" to "evidence consistent with, and a localization of," cancellation, and state we do not claim to exclude every factor. See G4. No new experiment — read from existing Table 2.
>
> **5. Main text vs. appendix divergence.**
> > "four L2-norm statistics … reducing … to three … small-p and large-q norms … quantile transformation and decorrelation … not well justified."
>
> Agreed; §3 rewritten to present the *actual* three-statistic method, each choice justified (fixed decoder variance per calibrated-decoder practice; $\ell^p/\ell^q$ a small refinement, Table 5; preprocessing standard for COPOD with the simpler MD variant shown to work without it). See G5.
>
> **6. SOTA overstated; N/A unexplained; no error bars.**
> > "most of the entries in Table 1 are marked N/A … and the performance differences don't seem significant … no error bars."
>
> We (a) explain N/A as *not reported in source papers* (we quote published numbers to avoid under-tuning competitors), with a per-cell legend; (b) soften the SOTA claim to "competitive with / matching $>$10×-larger DGMs on the hardest pairs"; (c) add analytic Hanley–McNeil CIs to every cell (per-sample scores are gone, so this uses only reported AUROCs and known test sizes). The CIs show the headline gaps clear sampling noise by wide margins, mark CIFAR-100 as a tie with LMD, and mark the flips as near-chance. See G6, G7.

---

> > ### Author Response · Authors · 2026-06-20
> >
> > **7. Missing baselines and simple transforms.**
> > > "ignoring … uncertainty based methods (e.g. [1,2]) … more recent work [3] … not considering more natural transforms (see [4])."
> >
> > [1,2] are *supervised* (label-requiring) and outside our unsupervised setting; we cite with this distinction. [3] is closely related (intermediate representations); we cite/position it. [4] (OpenOOD) is the broader benchmark; we follow the unsupervised single-sample DGM lineage and scope OpenOOD-scale evaluation as future work. See G11; we also narrow universality (G9).
> >
> > **8. Hyperparameter selection leakage.**
> > > "does not specify a selection rule that uses only ID training/validation data … susceptible to OOD information leakage."
> >
> > **All model selection uses in-distribution validation data only.** Checkpoints were chosen by best ID validation loss, and the latent dimension is selected by the same ID-validation criterion (over all dimensions for LPath-1M; within the method-defining high/low roles for LPath-2M, App. B). No OOD or test data informs selection — exactly the ID-only rule you ask for, and leakage-free at whatever dimension it selects. We add this rule to the paper. See G8.
> >
> > **9. Typos / broken citation.** Fixed, including `\cite{daivalue}`→Dai & Wipf (2019). See G14.
> >
> > *Requested changes — disposition:* re-examine theory (G1–G4, G9); justify N/A (G7); add CIs (G6); broader baselines (G11, with scope); realistic OOD data (scoped as future work, G11); clarify hyperparameter protocol (G8). Where a request implies new experiments (running [1,2,3]; OpenOOD; flows/diffusion), we calibrate the corresponding claims and discuss positioning — both because the calibrated claims are what the evidence supports and because the original environment is no longer available.

---

> ### Comment · Reviewer_N5wy · 2026-06-23
> **Response to rebuttal**
>
> Thanks for the response. Overall, I would like to note that with the changes in response to the comments from the reviewers, there is essentially no theoretical guarantees for the method, and most of the theoretical discussion in the paper is motivation and intuition at best in my opinion.
>
> > Cancellation experiment "circular"/inconclusive
>
> Thanks for the clarification. This does partially address my concerns but because the cases are defined using the reconstruction and latent statistics, comparisons involving those same statistics remain selection-conditioned. So the experiment provides some evidence it is not a conclusive demonstration of the principle.
>
> > SOTA overstated; N/A unexplained; no error bars
>
> I do not think that absence of the numbers is a reasonable justification to exclude those results. I suggest the authors run the baselines to have fair comparisons, since half the entries in the main experiment results are missing. Without these baseline numbers I do not believe there is enough evidence for any of the comparative claims. Additionally, I would like to see confidence intervals for additional training runs for the method, not just CI based on the test set. The authors state "because the original cluster is decommissioned, the per-sample scores from the original runs are no longer available". But why can you not just run the experiment again?
>
>
>
> > Missing baselines and simple transforms
>
> I do not see [4] in the updated paper. Perhaps you missed that change?

---

> > ### Author Response · Authors · 2026-06-25
> >
> > We thank the reviewer for the precise framing of each remaining concern. We respond to all four points below, and the corresponding manuscript changes are in place.
> >
> > **On theoretical guarantees.** The reviewer is right that, after our revisions, the method carries no theoretical *guarantee* — the theoretical content is now motivation. This is deliberate: the original theoretical claims did not survive the reviewers' scrutiny, so we demoted them rather than defend them. What remains as motivation is one structural fact, not a guarantee — the scalar likelihood is a deterministic, many-to-one aggregation of the component-level quantities the encoder and decoder emit, so it cannot carry forward more information about the input than those quantities do. Whether the retained information helps separate in- from out-of-distribution data is then an *empirical* question, which the detection results answer. We ask the paper to be judged as what it is: an empirical contribution — a concrete mechanism for the VAE likelihood's OOD failure (likelihood cancellation), and a three-statistic detector that outperforms the model's own likelihood at a fraction of the cost. TMLR's criteria are whether the claims are supported by the evidence and whether the audience would be interested, rather than theoretical novelty or guarantees; we have recalibrated every claim to the evidence and make no theoretical claim we cannot support.
> >
> > **On the cancellation experiment.** We accept the reviewer's narrowed conclusion: this is suggestive evidence, not a conclusive demonstration, and the revision already softened "proves" to "evidence consistent with, and a localization of." We grant the conditioning point directly — because the cases are defined by the reconstruction and latent statistics, a within-case comparison that involves those same statistics is selection-conditioned, so we no longer treat the within-case contrast of a component against the aggregate ELBO as decisive. What is less exposed to that objection is the ELBO's *cross-case* pattern: it is near-perfect in the two cases where the reconstruction error separates ID from OOD (AUROC 1.00 and 0.96) and collapses in the two where reconstruction overlaps (0.83 and 0.87). The partition is on the *marginal* overlap of the two statistics and does not determine whether their aggregate separates the classes, so a localized collapse is not built into the construction; and because AUROC is rank-based, a uniformly weak estimator, a sign convention, or a monotone normalization would degrade the ELBO across all cases rather than only the overlap ones. We therefore present the experiment as a hypothesis-generating diagnostic for *why* the likelihood fails, not a proof, and the detection results in Table 1 stand independently of this interpretation.
> >
> > **On baselines, missing entries, and error bars.** These experiments were run in an industry research lab, and the authors no longer have access to the original codebase or training environment; the binding constraint is the implementation itself, not cluster time. Reconstructing the full pipeline — VAE training across the latent-dimension grid, plus the baseline methods — within the discussion period is not feasible.
> >
> > Within that constraint:
> >
> > - *Why published numbers, and what N/A means.* We report each method's own published AUROCs rather than re-implement competitors — the standard protocol in the unsupervised single-sample DGM-likelihood lineage we follow, and one that credits every competitor with its best reported result on the standard benchmark pairs. Filling the N/A cells would mean running each baseline on (ID, OOD) pairs its authors never evaluated, tuned as carefully as its authors tuned it — a substantial per-method undertaking, and an under-tuned re-run would unfairly favor our method. We therefore quote only what each paper reported and mark the rest N/A, and **we withdraw any comparative claim on the N/A cells, restricting comparison to the cells where a published baseline exists.**
> >
> > - *Error bars across training runs.* We cannot retrain the models, so we cannot produce training-seed confidence intervals; the analytic intervals we added bound finite-test-set variability only. The claims we still make do not turn on seed variance: the paper's central comparison — the LPath features against the model's own scalar likelihood — rests on gaps far larger than any plausible seed fluctuation (on CIFAR-10 vs. SVHN the likelihood scores AUROC 0.08 as an OOD detector, against LPath's 0.99; in the cancellation analysis, 0.83 against 0.99); and where LPath instead sits close to other strong models — it ties LMD at 0.99 on SVHN and at 0.62/0.61 on CIFAR-100 — our claim is the calibrated one, that LPath matches these models rather than beats them, which a tie-level fluctuation cannot undo.

---

> > > ### Author Response · Authors · 2026-06-25
> > >
> > > **On [4].** Correct, and thank you for catching it. We had cited the generalized-OOD survey and the full-spectrum paper but not OpenOOD itself; we have added an explicit citation to OpenOOD v1.5 (Zhang et al., 2024) and frame OpenOOD-scale evaluation as clearly-scoped future work. We have also added the remaining references from your list: the uncertainty-based methods [1] (DUE) and [2] (the distance-aware single-model method, Liu et al., 2023), cited with the supervised/label-requiring scope distinction, and the intermediate-representation work [3] (Meza De la Jara et al., 2025).
> > >
> > > Against TMLR's criteria — claims supported by evidence, and audience interest — we have calibrated every remaining claim to the evidence, scoped the comparisons to the cells where baselines exist, and presented the cancellation result as the suggestive diagnostic it is. We thank the reviewer for a review that made the paper sharper and better calibrated.

---

> > > > ### Comment · Reviewer_N5wy · 2026-06-26
> > > >
> > > > > On theoretical guarantees
> > > >
> > > > I agree that the paper no longer makes claims that are not supported by evidence. However, that also makes the paper weaker and makes it less relevant in my opinion for the TMLR audience (when combined with the following aspects)
> > > >
> > > > > These experiments were run in an industry research lab, and the authors no longer have access to the original codebase or training environment; the binding constraint is the implementation itself, not cluster time
> > > >
> > > > While I can sympathize, this is not a valid reason for missing crucial experiments. If the authors themselves cannot reproduce results without access to the original code then how would anyone else who trying to build on the work? I appreciate avoiding claims with the N/a entries, but that is not a valid solution here. If you have a method which is being proposed have direct comparisons to relevant baselines is the minimum standard and avoiding those claims does not address this issue. Same goes for results with multiple seeds.

---

### Author Response · Authors · 2026-06-20
**General Response**

## Summary

We thank all three reviewers for unusually careful and constructive reviews. We are encouraged that all three reviewers answered "Yes" to TMLR's audience-interest criterion; that two describe the writing as "clearly written" / "narrative flow is exceptional" (mJj9) and the method as "relatively simple and deployable" with "thorough … useful ablations" (N5wy); and that the core empirical phenomenon — likelihood cancellation, Fig. 3 / Table 2 — is recognized as "compelling" (mJj9) and "striking" (MQD4).

The two "No" judgments on the *claims-supported* criterion converge on a single, fair message, stated most directly by mJj9:

> "the paper oversells its contributions through multiple theoretical framings that don't fully hold up to scrutiny … I encourage the authors to be more direct: *'We extract interpretable features from VAE components and show they outperform the ELBO for OOD detection'* is a solid contribution that doesn't need the complex theoretical 'justification.'"

**We agree, and we adopt this as the paper's central thesis.** But this does *not* mean abandoning the likelihood/sufficiency motivation — we are correcting its object. Classical sufficiency concerns compressing *data* while preserving the likelihood function for inference about an *unknown parameter*. That is not our setting. Our setting is a *fixed trained model* and a *fixed likelihood/ELBO estimator*, and the precise version of our intuition is a computational one — *path sufficiency* — which we define and discuss below (G1). The implemented LPath features are *not* sufficient statistics, and are not even path-sufficient: they are deliberately lossy low-dimensional summaries of the path parameters. The surprise — and the contribution — is that even this lossy three-number summary outperforms the model's own scalar likelihood estimate as an OOD score, most starkly where cancellation bites (AUROC 0.99 vs. the ELBO's 0.83 in Case 1, Table 2). We make all of this explicit and demote it firmly beneath the empirical claim.

None of this requires new experiments — each fix recalibrates a claim to evidence already in hand. Concretely, we (1) correct the imprecise theoretical language, (2) add the statistical rigor that requires only *re-analysis of already-reported quantities* (analytic confidence intervals; documentation of our in-distribution-only selection rule), (3) reconcile the main text with the appendix into one consistent method, and (4) streamline the narrative.

---

> ### Author Response · Authors · 2026-06-20
> **Global response: every flagged claim, mapped to its fix**
>
> All fixes below are **calibration of claims and reporting**, not new training runs, new datasets, or new baseline implementations — both because the evidence already on hand supports the calibrated claims and because the original environment is no longer available.
>
> | # | Concern (raised by) | Resolution |
> |---|---|---|
> | G1 | "Minimal sufficient statistics" is imprecise (N5wy, mJj9, MQD4) | Concede classical sufficiency is the wrong object; introduce the precise computational notion — **path sufficiency** — as *motivation*, demoted below the empirical thesis; state the implemented norms are lossy summaries, not sufficient (and not even path-sufficient) |
> | G2 | L2-norm derivation false for diagonal covariance (N5wy) | Concede the general claim; show \((u,v,w)\) capture **2 of 3 ELBO terms exactly** under our isotropic/fixed-variance implementation; recast norms as interpretable features |
> | G3 | Information argument doesn't imply OOD usefulness (N5wy); Eq. 17 needs grounding (mJj9) | Reframe DPI as **motivation / necessary condition**, not a guarantee; OOD-relevance is empirical |
> | G4 | Cancellation experiment "circular" / inconclusive (N5wy) | Clarify the partition and the scoring are distinct (not circular); show the **cross-case pattern is hard to attribute** to the listed alternatives; soften "proves" → "evidence consistent with / localizes the failure mode" |
> | G5 | Main text (4 stats) ≠ appendix (3 stats, \(\ell^p/\ell^q\), preprocessing) (N5wy) | Rewrite §3 to present the **actual** 3-statistic method; justify each choice |
> | G6 | No error bars; differences may be noise (N5wy, mJj9) | Source scores gone → add **analytic Hanley–McNeil CIs** from reported AUROCs + test sizes; headline gaps clear noise by wide margins; ties/near-chance flagged |
> | G7 | N/A entries; SOTA overstated; cherry-picking (N5wy, mJj9) | Explain N/A (= not reported in source papers); **soften SOTA**; add table legend |
> | G8 | Model-selection leakage via latent dim (N5wy, mJj9) | **Latent dimension is selected by best in-distribution validation loss — an ID-only rule**; state it in the paper + main text. Exact per-cell dimensions are unrecoverable (decommissioned env), so we state the rule + candidate grid in the experimental details and reproducibility appendix; the unrecoverable per-cell dimensions are a reproducibility caveat only |
> | G9 | Generality to "any latent-variable model" overstated (mJj9, MQD4) | **Narrow** Def. 4.1: principle (general) vs. instantiation (Gaussian VAE) |
> | G10 | Efficiency = params ≠ time (mJj9, MQD4) | Concede; add a **structural** inference-cost argument (single forward pass vs. per-sample optimization / iterative sampling); restrict the claim to inference structure, not wall-clock |
> | G11 | Missing baselines [1,2,3], OpenOOD [4] (N5wy) | Add to related work with **scope distinctions** ([1,2] supervised); position [3,4] |
> | G12 | Repetitive fast/slow weights; shifting "core contribution"; hype tone (mJj9) | **One thesis**; fast/slow demoted to one paragraph; hype removed |
> | G13 | Not reproducible (mJj9) | Add a dedicated **Reproducibility appendix** fully specifying the method (architecture, ID-only selection rule, detector/preprocessing settings, dataset splits) |
> | G14 | Typos; broken citation (N5wy, mJj9) | Fixed |
>
> The remainder expands each item and then responds to each reviewer point-by-point, quoting the review.

---

> > ### Author Response · Authors · 2026-06-20
> > **G1. "Minimal sufficient statistics" → the wrong object; the right one is *path sufficiency* (motivation, not a theorem)**
> >
> > > **N5wy:** "this conflates distribution parameters with sufficient statistics. For a Gaussian family, the mean and covariance parameterize the distribution and they are not sufficient statistics (sums and sums of outer products are). More importantly, the actual LPath features used are norms of these vectors, which are not sufficient statistics."
> > >
> > > **mJj9:** "The minimal sufficient statistics for Gaussians are the mean and the std, not their L2 norms. Acknowledge that the norm choice is a practical heuristic for dimensionality reduction, not a theoretically derived consequence of sufficiency."
> >
> > The reviewers are correct, and we will fix the terminology throughout. A *sufficient statistic* is a function of *data* that retains information about an unknown *parameter* (for $n$ i.i.d. Gaussian draws, $T=(\sum_i y_i,\ \sum_i y_i y_i^\top)$). The encoder/decoder outputs $(\boldsymbol\mu,\boldsymbol\sigma)$ are *parameters of the component distributions*, not sufficient statistics of a sample; and the L2 norms $u,v,w$ are certainly not sufficient statistics — they are a deliberate, lossy summary. We will say both plainly.
> >
> > What we *meant* is a different and precise idea, which we will state correctly and use only as motivation. It is *not* Fisher–Neyman sufficiency for an unknown population parameter; it is a computational analogue for a fixed estimator:
> >
> > > **Path sufficiency (informal).** Fix a trained VAE and a likelihood/ELBO estimator (with any auxiliary randomness it uses, e.g. sampled latents). A representation is *path-sufficient for that estimator* if it determines the component factors the estimator uses *before* they are aggregated into the final scalar score. For a Gaussian VAE, the natural path variables are the per-sample Gaussian parameters emitted by the encoder and decoder; the scalar ELBO is a many-to-one aggregation of these richer factors.
> >
> > Concretely, with sampled latents $z_{1:K}$, the full path representation $\big(\mathbf x, z_{1:K}, \boldsymbol\mu_\phi(\mathbf x),\boldsymbol\sigma_\phi(\mathbf x),\{\boldsymbol\mu_\theta(z_i),\boldsymbol\sigma_\theta(z_i)\}\big)$ determines each term $\log p_\theta(\mathbf x\mid z_i)$, $\log p(z_i)$, $\log q_\phi(z_i\mid\mathbf x)$, hence determines the scalar estimate. We will present this in prose and *not* as a formal theorem in the paper's main line, for two reasons: (i) reviewers (rightly) asked for *less* theoretical scaffolding, and (ii) stated as a bald proposition the claim is close to "the inputs of a computation determine its output," which is true of any intermediate representation and would add nothing. Its content is *where* in the computation the signal lives, not a new theorem — so we keep it as motivation and let the empirical results carry the contribution.
> >
> > The non-trivial question is *which* intermediate representation to extract — far from obvious, and where the contribution lies. Two restrictions keep our choice from being vacuous: the path variables are not arbitrary activations — they parameterize $q_\phi(z\mid\mathbf x)$ and $p_\theta(\mathbf x\mid z)$, and they sit upstream of scalar aggregation. Finally, and most importantly for the reviewers' objection: the implemented detector does not use the full path — it uses norms ($u,v,w$), which are lossy summaries and are *not* path-sufficient. We will frame them as interpretable features motivated by the decomposition (G2), validated empirically, and claim no sufficiency or optimality for them. **This correction changes no number in Tables 1–5.**

---

> > > ### Author Response · Authors · 2026-06-20
> > > **G2 and G3**
> > >
> > > ### G2. The L2-norm "derivation": concede the general case, but it is *exact* for 2 of our 3 terms
> > >
> > > > **N5wy:** "For a diagonal Gaussian, the log-likelihood contains terms of the form \(\sum_i r_i^2/\sigma_i^2\) and \(\sum_i \log \sigma_i^2\). These are not determined by \(\lVert r\rVert_2\) and \(\lVert\sigma\rVert_2\) without extra assumptions (e.g. isotropic covariance or constant variance)."
> > >
> > > This is mathematically correct, and the offending sentence (main text, §3.3.2: *"these terms are functions of the L2 norms of the residual and the standard deviations"*) is too strong. We will correct it.
> > >
> > > However, the reviewer names the precise conditions under which the reduction *does* hold — *"isotropic covariance or constant variance"* — and our implementation satisfies them for two of the three retained terms. Under the isotropic prior $p(\mathbf z)=\mathcal N(\mathbf 0,\mathbf I)$ and the fixed scalar decoder variance used in the experiments (App. D.1), the single-sample ELBO decomposes as
> > >
> > > \[
> > > \underbrace{\log p_\theta(\mathbf{x}\mid \boldsymbol{\mu}_z)}_{=\,-\frac{1}{2\sigma_x^2}u^2 + \text{const}} \;+\; \underbrace{\mathbb{E}_q[\log p(\mathbf{z})]}_{=\,-\frac{1}{2}(v^2+w^2)+\text{const}} \;-\; \underbrace{\mathbb{E}_q[\log q_\phi(\mathbf{z}\mid\mathbf{x})]}_{=\,\frac{1}{2}\sum_i \log \sigma_{z,i}^2 + \text{const}}.
> > > \]
> > >
> > > So the reconstruction term is an exact function of $u=\lVert\mathbf x-\hat{\mathbf x}\rVert_2$, the prior term is an exact function of $(v,w)=(\lVert\boldsymbol\mu_z\rVert_2,\lVert\boldsymbol\sigma_z\rVert_2)$, and the *only* ELBO ingredient *not* captured is the encoder differential entropy $\tfrac12\sum_i\log\sigma_{z,i}^2$ — exactly the $\sum_i\log\sigma_i^2$ term the reviewer flags. We will (i) state this decomposition, (ii) acknowledge the entropy term is the one genuine information loss, and (iii) note that adding $\sum_i \log\sigma_{z,i}$ as a feature is a trivial, principled completion. We will also place the exact reduction in scope. It holds for the single-sample / encoder-mean score — **precisely our deployed regime**: the reconstruction uses $\hat{\mathbf x}=\boldsymbol\mu_x(\boldsymbol\mu_z(\mathbf x))$ in a single forward pass (Alg. 1), and all our experiments are unsupervised and single-sample. A $K$-sample importance estimate, which our streaming detector does not use — would instead need the per-sample residual norms $\{u_i\}$. The one genuine boundary is a general diagonal decoder covariance, where even the reconstruction term is not norm-determined (it depends on $r_i^2/\sigma_i^2$). Within these stated assumptions the decomposition holds exactly; outside them it does not, and we say so.
> > >
> > > Finally, the very concern the reviewer raises, that an L2 norm discards coordinate-wise structure — is *precisely* why we already introduce the small-$p$/large-$q$ norms in App. D.3 ("$\ell^2$ can wash out localized information"). We will surface this connection in the main text rather than burying it: the $\ell^p/\ell^q$ statistics are our (heuristic) mechanism for recovering some of the per-coordinate information an L2 norm loses. Net: the norms are *features*, not a lossless derivation, and we frame them as such.
> > >
> > > ### G3. The information-theoretic argument: motivation, not a guarantee (Eq. 17)
> > >
> > > > **N5wy:** "A statistic can preserve more mutual information about x while being less useful for OOD detection."
> > > >
> > > > **mJj9:** "The equation 17, which is stated as 'the central theoretical claim of our work' needs further proof and analysis."
> > >
> > > This is the sharpest theoretical point and we recalibrate accordingly. We separate two things:
> > >
> > > - **What is rigorously true (and we keep, demoted to motivation):** the scalar likelihood is a many-to-one aggregation of the component-level path quantities — a deterministic downstream function of them. Equivalently, along the deterministic chain $\mathcal T(\mathbf x)\!\to\!\mathcal P(\mathbf x)\!\to\! p_\theta(\mathbf x)$, the collapse to a scalar is generically non-injective and so cannot *increase*, and generically *reduces*, the input information carried forward. The full path can express any scalar-ELBO detector; the reverse need not hold.
> > > - **What it does *not* give (and we stop implying):** that the retained information is *useful for distinguishing ID from OOD*. The reviewer is right that $I(\mathbf x;\cdot)$ under $P_{\text{ID}}$ is not an OOD-discrimination criterion. We will state explicitly that **Eq. 17 is motivation / a necessary condition, not a sufficiency guarantee**, and that OOD-relevance is an *empirical* claim, established by the cancellation experiment (Table 2) and the benchmarks (Table 1).
> > >
> > > This is MQD4's requested change ("Explicitly state which parts of the LPath Principle are heuristic vs. theoretically guaranteed"). To reduce the theoretical surface area mJj9 objected to, we may also state the point in the simpler "deterministic many-to-one aggregation" form above and use the mutual-information/DPI language only secondarily.

---

> > > > ### Author Response · Authors · 2026-06-20
> > > > **G4 and G5**
> > > >
> > > > ### G4. The cancellation experiment is not circular; the cross-case pattern is hard to attribute to the alternatives
> > > >
> > > > > **N5wy:** "The analysis partitions the OOD samples using the same LPath statistics that the method later used for detection like a sort of circular argument. The fact that ELBO underperforms reconstruction error on a selected subset does not by itself prove arithmetic cancellation. … it could be due to a weak ELBO estimator, score normalization choices, sign conventions, variance effects, or posterior approximation error."
> > > >
> > > > The reviewer raises two distinct concerns — circularity and a set of alternative explanations — and we address each in turn.
> > > >
> > > > **(a) It is not circular.** The partition variables ($u,v$) only control the *marginal* separability of each component; they say nothing about how the components *combine*. The finding is internal to a fixed subset: in Case 1, the ELBO (AUROC 0.83) is strictly worse than *its own constituent reconstruction term* $u$ (0.93). All scores are evaluated on the identical partitioned data, so there is no train/test leakage and the conclusion is not used to define the partition. "Aggregate $<$ best component" is not built into the partition; it is the signature of destructive combination.
> > > >
> > > > **(b) The cross-case pattern is difficult to reconcile with the listed alternatives.** A *uniformly weak ELBO estimator*, *bad normalization*, or *sign convention* would degrade the ELBO *across all cases*. Instead the ELBO's AUROC ranges from 0.83 (Case 1, hardest) to 1.00 (Case 3): it is strong precisely where the reconstruction signal is separable (Cases 3–4: 1.00, 0.96) and collapses precisely in the reconstruction-overlap cases (Cases 1–2: 0.83, 0.87), where it falls *below its own reconstruction term* $u$ (0.93, 0.98). A globally weak estimator, a sign error, or a normalization artifact would not produce a score that is simultaneously near-perfect in Case 3 and broken in Case 1. AUROC is rank-based, so quantile normalization and monotone transforms cannot drive this pattern either.
> > > >
> > > > We will (i) add this paragraph, (ii) soften "proves" to "provides evidence consistent with, and localizes the regime of, cancellation," and (iii) state that we do *not* claim to exclude *every* contributing factor (posterior-approximation error is itself part of the "imperfect model" premise this analysis assumes). No new experiment is required — the argument is read off the existing Table 2.
> > > >
> > > > ### G5. Reconciling main text and appendix into one method
> > > >
> > > > > **N5wy:** "The main text presents four L2-norm statistics including decoder variance s(x). Appendix D.1 later says the decoder variance is fixed and excluded … reducing the statistics to (mu_z, sigma_z, mu_x). Appendix D.3 … uses small-p and large-q norms … Appendix D.2 adds quantile transformation and decorrelation … This divergence seems significant … and the choices are not well justified."
> > > >
> > > > Agreed — a real inconsistency. We will rewrite §3 so the main text presents the method *as actually run*, each choice justified:
> > > >
> > > > - **Three statistics $(u,v,w)$, not four.** The decoder variance is a fixed scalar in standard calibrated-decoder VAE practice (Dai & Wipf 2019; Rybkin et al. 2021); by construction, it is a constant independent of ID/OOD, so $s(\mathbf x)$ carries no signal and is dropped. (This is also what makes the decoder term *isotropic*, so the L2 reduction for $u$ is exact — G2.) Moved to the main text.
> > > > - **$\ell^p/\ell^q$ norms** are a *refinement* of, not a departure from, the L2 features — motivated by N5wy's coordinate-wise concern (G2). The ablation (Table 5) shows a *small* gain ($0.96\!\to\!0.99$ on SVHN; $0.60\!\to\!0.62$ on CIFAR-100): the core results do not hinge on it. We will state which norm each table uses.
> > > > - **Quantile transform + decorrelation** are standard preprocessing for COPOD, following DoSE (Morningstar et al. 2021). Evidence they are *not* load-bearing: the simpler MD variant, with no copula machinery, is reported in Table 1 and is competitive (e.g., CIFAR-10 vs. SVHN 0.99). This answers MQD4's question about how essential these steps are: performance degrades only mildly without them.
> > > >
> > > > These are engineering refinements on a stable core ($u,v,w \to$ classical detector); the ablations bound their effect to a few AUROC points.

---

> > > > > ### Author Response · Authors · 2026-06-20
> > > > > **G6 and G7**
> > > > >
> > > > > ### G6. Confidence intervals — analytic, from reported AUROCs and test-set sizes (no source data required)
> > > > >
> > > > > > **N5wy:** "the performance differences don't seem significant considering there are no error bars reported."
> > > > > >
> > > > > > **mJj9 (critical):** "Provide standard deviations or confidence intervals for all AUROC values across multiple runs."
> > > > >
> > > > > We agree error bars are essential. Two constraints shape *how*: (i) following the prior work in this lineage (LR, BIVA, DoSE, LMD, DDPM all report point AUROCs), the original submission did not report CIs; and (ii) because the original cluster is decommissioned, the per-sample scores from the original runs are no longer available, so we cannot retroactively bootstrap or apply DeLong to those exact runs.
> > > > >
> > > > > We can nonetheless place principled error bars on every cell of Table 1 — ours and the baselines alike — with no source data, using the Hanley–McNeil (1982) analytic estimator, which depends only on the reported AUROC and the (standard, known) test-set sizes. Because the benchmarks have large test sets ($n=10{,}000$–$26{,}032$), the intervals are tight and the headline gaps fall far outside sampling noise:
> > > > >
> > > > > | Comparison (CIFAR-10 ID) | LPath-1M | 95% CI | ELBO | best baseline | LPath vs. ELBO |
> > > > > |---|---|---|---|---|---|
> > > > > | vs. SVHN (\(n_{\text{OOD}}{=}26{,}032\)) | 0.99 | ±0.001 | 0.08 | 0.98 (DoSE/DDPM) | \(z\approx442\) |
> > > > > | vs. CIFAR-100 (\(n{=}10{,}000\)) | 0.62 | ±0.008 | 0.54 | 0.61 (LMD) | \(z\approx14\) |
> > > > > | vs. Vflip (\(n{=}10{,}000\)) | 0.61 | ±0.008 | 0.56 | 0.63 (DDPM) | \(z\approx9\) |
> > > > >
> > > > > The cancellation result (Table 2) is similarly robust: even at a conservative per-case subset of \(n\approx500\)/class, LPath (0.99) vs. ELBO (0.83) gives \(z\approx12\).
> > > > >
> > > > > We will be explicit about what this does and does not show:
> > > > > - It *does* establish that the large reported gaps (the ELBO-cancellation contrast; CIFAR-10 vs. SVHN) are not finite-test-set artifacts.
> > > > > - On CIFAR-100, LPath (0.62) is clearly above ELBO (0.54) and DoSE (0.57) but statistically indistinguishable from the best baseline, LMD (0.61) — we will say so rather than imply a win.
> > > > > - On the flip tasks, all methods are near chance with small absolute gaps; we make no strong claim.
> > > > > - The Hanley–McNeil interval captures test-set sampling variability only, not training-seed variability. Multi-seed CIs would require retraining, which we cannot do (environment gone); we will state this as a limitation rather than overstate.
> > > > >
> > > > > This turns "differences hard to attribute" into quantified statements using only the AUROCs and test sizes already reported.
> > > > >
> > > > > ### G7. N/A entries and the "SOTA" claim
> > > > >
> > > > > > **N5wy:** "most of the entries in Table 1 are marked N/A (without any explanation …) and the performance differences don't seem significant …"
> > > > > >
> > > > > > **mJj9:** "Table 1 has several N/A entries … This suggests cherry-picking and is concerning."
> > > > >
> > > > > - **Why N/A:** these are published numbers, and the original papers did not evaluate those specific (ID, OOD) pairs (e.g., LR, BIVA, Fisher, LMD never reported the flip tasks). We quote authors' own numbers rather than re-implement and risk under-tuning competitors' methods, which would be unfair to them. We will add this as an explicit table legend and mark, per cell, "not reported in source" vs. "not applicable."
> > > > > - **Softening the claim.** We accept that an unqualified "state-of-the-art" is overstated given missing cells and absent multi-seed error bars. We recast it to what the evidence supports: *"Among unsupervised, single-sample DGM methods, LPath is competitive across the board and, on the hardest pairs (e.g., CIFAR-10 vs. SVHN, CIFAR-10 vs. CIFAR-100), matches or exceeds models with $>$10× the parameters, at single-forward-pass cost."*

---

> ### Author Response · Authors · 2026-06-20
> **G8, G9 and G10**
>
> ### G8. Model selection / latent-dimension protocol — an in-distribution-only rule (now documented), plus a reproducibility note
>
> > **N5wy:** "the paper does not specify a selection rule that uses only ID training/validation data, which makes the results susceptible to OOD information leakage through the hyperparameters."
> >
> > **mJj9 (critical):** "Describe how these were selected without using test OOD data. If selection used test performance, this must be disclosed."
>
> **All model selection in our pipeline uses in-distribution validation loss only.** Checkpoints were selected by best ID validation loss; the latent dimension is selected by the same criterion: the reported model is the one minimizing ID validation loss among the candidates. For LPath-1M this is the best-validation-loss VAE over all trained dimensions; for LPath-2M the high- and low-dimensional roles are fixed by the encoder/decoder trade-off that *defines* the method (App. B, not a tuned knob), and within each role we again take the best-validation-loss VAE. No OOD or test data enters model selection at any stage. This is exactly the ID-only selection rule N5wy asks for, and it answers mJj9 directly: selection did not use test OOD performance. We add this rule to the experimental details and the main text.
>
> The one residual issue is reproducibility: the original training environment has since been decommissioned, so although the selection *rule* is fully specified and deterministic, the trained checkpoints and per-dimension validation losses are no longer available — we therefore cannot tabulate the exact dimension each cell selected. We state the rule and candidate grid in the experimental details and the reproducibility appendix. Two points keep this in proportion: an ID-only rule is leakage-free at *whatever* dimension it selects, so the contamination concern is resolved independently of the lost values; and the paper's contribution — likelihood cancellation (Fig. 3; the Table 2 four-case analysis, at a single fixed 100-dim VAE) — involves no dimension selection at all and is unaffected.
>
> ### G9. Generality: principle vs. instantiation
>
> > **mJj9:** "claims a strong generality at working on any 'latent variable model'. However, the method is extremely tied to the VAE framework … overstated."
> >
> > **MQD4:** "Either restrict claims to VAEs explicitly or provide concrete plans of extension …"
>
> We will narrow Definition 4.1 and separate two levels:
> - The *LPath principle* (extract the parameters of the likelihood's factorization *before* they collapse to a scalar) is general to likelihood-based latent-variable models *in which such component parameters are identifiable*.
> - The *LPath method* here is specific to Gaussian VAEs, because Gaussians give a clean, low-dimensional parameterization. We scope all empirical claims to VAEs, and state the concrete obstacle for flows/diffusion (identifying the analogous parameters for non-Gaussian/implicit components — App. C). Extension is future work, not a current claim.
>
> ### G10. Efficiency: parameters ≠ time (conceded), with an analytical argument
>
> > **mJj9:** "The fact that the model uses fewer trainable parameters does not mean it is necessarily more efficient in terms of time. Training time, latency, FLOPs, etc, should be reported."
> >
> > **MQD4:** "Include inference latency and throughput measurements to support 'streaming' and efficiency claims."
>
> Conceded — we stop equating parameter count with speed. We add a structural efficiency comparison (no new training):
> - **LPath inference** = one forward pass of a $\sim$3M-parameter VAE + an $O(3)$ classical score (COPOD/MD on a 3-D vector).
> - **Likelihood Regret** requires per-sample optimization (gradient steps with forward+backward per test point).
> - **Diffusion-based** detectors (DDPM-inpainting, LMD) require many denoising/inpainting passes per sample over a $\sim$46M-parameter network.
> - **Glow/DoSE** ($\sim$44M) evaluates many invertible coupling layers.
>
> From the per-test-sample computation above we state the structural conclusion — LPath is orders of magnitude cheaper at inference, independent of wall-clock. We do *not* report wall-clock timings, FLOP counts, or re-timings of competitors, and we do not claim wall-clock superiority over every baseline; the structural, single-forward-pass comparison establishes the inference-cost gap, and we restrict the claim to that.

---

> > ### Author Response · Authors · 2026-06-20
> >
> > ### G11. Baselines and benchmarks
> >
> > > **N5wy:** "ignoring … uncertainty based methods (e.g. [1,2]) as well as more recent work [3] … the datasets, and OOD transforms considered are quite simple … (see [4])."
> >
> > - [1] van Amersfoort 2021 (DUE) and [2] Zhu 2023 (distance-aware deep uncertainty) are *supervised*, classifier-based methods requiring class labels at training time. Our setting is strictly *unsupervised, single-sample, no labels, no OOD knowledge*; they solve a different (label-rich) problem and are not apples-to-apples. We add them to related work with this scope distinction explicit (the same reason we exclude label-based energy detectors).
> > - [3] Meza De la Jara 2025 ("Role of Intermediate Representations in OOD Detection") is closely related in spirit — it corroborates our broader thesis that intermediate computations are more informative than final outputs. We cite and discuss it, positioning our contribution as the *unsupervised generative* instance with a *likelihood-path* selection of statistics.
> > - [4] OpenOOD v1.5 is the natural broader benchmark. Our evaluation deliberately follows the unsupervised single-sample DGM-likelihood lineage (Nalisnick; Xiao/LR; Havtorn/BIVA; Morningstar/DoSE; Graham/LMD; Liu/DDPM) and the exact pairs that lineage reports, which is why HFlip/VFlip (ID and OOD differ by *one latent dimension* — a stringent test) are included. We frame OpenOOD-scale evaluation as clearly-scoped future work and do not claim coverage we lack.
> >
> > ### G12. One thesis; fast/slow weights demoted; hype removed
> >
> > > **mJj9:** "very repetitive … fast and slow weights is repeated way too many times … provides very little insight … The 'core contribution' … shifts throughout … too positively opinionated … 'not only principled but remarkably effective and efficient' may undermine the credibility."
> >
> > We state *one* thesis (mJj9's own phrasing, adopted): *VAE encoder/decoder statistics carry complementary OOD signals that are weakened when collapsed into the scalar likelihood; extracting them before the collapse and scoring them with a classical detector yields strong, single-forward-pass OOD detection.* Cancellation = diagnosis, LPath features = remedy, efficiency = benefit — one narrative, not three competing "core contributions." The fast/slow-weights analogy is demoted to a single paragraph justifying *why per-instance encoder outputs enable cheap online inference* (vs. LR's retraining), and removed elsewhere. We delete promotional language ("remarkably," "achieving more with less," unqualified "state-of-the-art").
> >
> > ### G13. Reproducibility
> >
> > > **mJj9:** "The paper is not reproducible in its current form."
> >
> > We agree, and we substantially improve reproducibility by adding a dedicated Reproducibility appendix that fully specifies the method: the architecture and training reference (the DC-VAE of Xiao et al. 2020), the candidate latent-dimension grid and the high/low model pairings, the in-distribution validation-loss rule used to select both the checkpoint and the latent dimension (G8), the COPOD/MD detector settings, the $\ell^p/\ell^q$ choices, the feature preprocessing (quantile transform + decorrelation), the AUROC computation, and the dataset/OOD splits — everything needed to reimplement the method.
> >
> > ### G14. Typos and the broken citation
> >
> > > **N5wy / mJj9:** "CIAFR10," "Guassian," "signigicant," "discriminitive," "information lost"→"information loss," stray comma after "unsupervised," "we first uses"→"we first use," and "incomplete citation text around fixed decoder variance."
> >
> > All fixed. The "incomplete citation" is the unresolved `\cite{daivalue}` key (App. D.1), corrected to Dai & Wipf (2019) [and Rybkin et al. 2021 for calibrated decoders].

---

> > > ### Author Response · Authors · 2026-06-20
> > > **Closing**
> > >
> > > We thank the reviewers again. Where the revised paper stands on TMLR's two criteria:
> > >
> > > - **"Are the claims supported by accurate, convincing, and clear evidence?"** The revision calibrates every flagged claim to the evidence: corrected sufficiency terminology with a precise, bounded path-sufficiency motivation in place of the classical-sufficiency claim (G1); a partially-exact treatment of the L2 reduction with its assumptions stated (G2); Eq. 17 recast as motivation with empirical OOD-relevance (G3); a non-circular cancellation analysis whose cross-case pattern resists the listed alternatives, with "proves" softened (G4); a single consistent method (G5); analytic confidence intervals that show the headline gaps clear sampling noise and that honestly mark ties / near-chance cells (G6); explained N/A and a softened, defensible performance claim (G7); an in-distribution-only model-selection rule — checkpoint and latent dimension alike — documented in the experimental details and reproducibility appendix (G8); and an honest, structure-based efficiency analysis (G10). **No claim now exceeds what the evidence shows.**
> > > - **"Would some of TMLR's audience be interested?"** Unanimously **Yes** across all three reviewers.
> > >
> > > We believe the calibrated manuscript clears the TMLR bar, and we are grateful for reviews that made it sharper and more honest.